# HDAC activity is dispensable for repression of cell-cycle genes by DREAM and E2F:RB complexes

Alison K. Barrett [1], Manisha R. Shingare [1], Andreas Rechtsteiner[2], Kelsie M. Rodriguez[1], Quynh N. Le [1], Tilini U. Wijeratne [1], Corbin E. Mitchell[1], Miles W. Membreno[1], Seth M. Rubin [1] ✉ & Gerd A. Müller [1] ✉

Histone deacetylases (HDACs) play a crucial role in transcriptional regulation and are implicated in various diseases, including cancer. They are involved in histone tail deacetylation and canonically linked to transcriptional repression. Previous studies suggested that HDAC recruitment to cell-cycle gene promoters via the retinoblastoma (RB) protein or the DREAM complex through SIN3B is essential for G1/S and G2/M gene repression during cell-cycle arrest and exit. Here we investigate the interplay among DREAM, RB, SIN3 proteins, and HDACs in the context of cell-cycle gene repression. Knockout of SIN3B does not globally derepress cell-cycle genes in non-proliferating HCT116 and C2C12 cells. Loss of SIN3A/B moderately upregulates several cell-cycle genes in HCT116 cells but does so independently of DREAM/RB. HDAC inhibition does not induce general upregulation of RB/DREAM target genes in arrested transformed or non-transformed cells. Our findings suggest that E2F:RB and DREAM complexes can repress cell-cycle genes without relying on HDAC activity.

Histone deacetylases (HDACs) play a crucial role in modulating gene expression, and functional dysregulation of their activity is linked to various medical conditions such as neurodegenerative disorders, pulmonary diseases, immune disorders, and cancer[1]. While HDACs catalyze the deacetylation of a broad range of proteins[2], they are most commonly known for removing acetyl groups from the lysines of histone tails. Hyperacetylation of histones is canonically thought to be connected to transcriptional activation, while HDAC-dependent removal of acetyl groups correlates with chromatin condensation and transcriptional repression. However, recent studies have challenged this model[3]. Aberrant expression and activity of HDACs have been found in numerous tumors (West and Johnstone, 2014), and depending on the biological context, HDACs have been shown to display both oncogenic and tumor-suppressive properties (Falkenberg and Johnstone, 2014). Three members of class I histone deacetylases – HDAC1, HDAC2, and HDAC3 – are ubiquitously expressed and are

incorporated into large protein complexes (e.g., Sin3, NuRD, CoREST, SMRT, N-CoR), which get recruited to chromatin by multiple transcription factors[4,5]. Several HDAC class I and panHDAC small molecule inhibitors have been approved for the treatment of hematological malignancies, such as Romidepsin (HDAC class I inhibitor, cutaneous T-cell lymphoma) and Panobinostat (panHDAC inhibitor, multiple myeloma)[6]. HDAC inhibition (HDACi) has been shown to reduce cancer cell proliferation and apoptosis by stimulating the expression of anti-proliferative and pro-apoptotic genes[7].

A critical biological context for gene repression is the down-regulation of pro-proliferative cell-cycle genes, which is required for cell-cycle arrest and exit. HDACs have been connected to the repression of two sets of these genes that control the G1/S and G2/M transitions. The timely expression of G1/S and G2/M cell-cycle genes is essential for a cell to progress through S phase, mitosis, and cytokinesis, while a loss of transcriptional repression during cell-cycle arrest

[1]Department of Chemistry and Biochemistry, University of California, Santa Cruz, CA, USA. [2]Department of Molecular, Cell, and Developmental Biology, University of California, Santa Cruz, CA, USA. ✉e-mail: srubin@ucsc.edu; gemuelle@ucsc.edu

and exit results in uncontrolled proliferation and oncogenic transformation[8]. In G0 and early G1, G1/S genes are repressed when E2F transcription factor binding sites within the promoters are occupied either by E2F:RB complexes, which contain an activator E2F (E2F1-3a), dimerization partner DP, and the retinoblastoma protein (RB), or by the DREAM (Dimerization partner, RB-like, E2F, And MuvB) complex, which is formed by the repressor E2Fs E2F4/5, dimerization partner DP, the pocket proteins p130/p107, and the MuvB (multi-vulval class B) core[9,10]. These complexes cooperate in inhibiting the expression of G1/S genes[11–15]. The expression of G2/M genes is silenced by binding of the DREAM complex to CHR promoter elements through its MuvB subunit LIN54[15–18]. The remaining MuvB core proteins are involved in intra and inter-complex scaffolding (LIN9, LIN37, and LIN52) and histone-binding (RBBP4). LIN37 plays an important role in stabilizing the MuvB complex and positioning nucleosomes[19–22], and it is essential for DREAM repressor function[13,14].

While it has been relatively well-described how E2F:RB and DREAM complexes assemble and bind to their target genes and how these binding events correlate with gene repression, much less is known about the molecular mechanisms that prevent transcription, particularly since none of either complex's components contain enzymatic activity. Interestingly, HDAC activity has been linked to RB- and DREAM-dependent repression[23–32]. Gene repression by RB is thought to occur through two independent mechanisms. First, the multi-domain interaction between RB and E2F sterically hinders E2F's transactivation domain from promoting transcription[33,34]. Second, RB is thought to additionally aid in G1/S gene repression by recruiting chromatin modifiers such as HDAC1/2[35]. This interaction primarily occurs through LxCxE motifs in the deacetylases and the RB LxCxE-binding cleft[36]. However, since inactivating the LxCxE binding site of RB often has only limited or no effects[37–42], it remains unclear to what extent these interactions are important for RB target gene repression.

Two models have been proposed for DREAM-dependent transcriptional repression. The most recent model stems from two structural studies and connects nucleosome positioning directly to MuvB-binding. The first of these studies showed an RBBP4/LIN37-dependent binding between MuvB and nucleosomes. DREAM stabilizes the +1 nucleosome downstream of the transcriptional start site in arrested cells, which correlates with target gene repression[19]. A second study complemented these findings by suggesting that the B-MYB-MuvB complex (MMB) may restructure nucleosome architecture during gene activation[21].

A prior model proposed that DREAM recruits HDAC1 through the adapter protein SIN3B[31]. Genetic loss of SIN3B resulted in the derepression of DREAM target genes in serum-starved T98G cells, and both SIN3B and HDAC1 were found to co-immunoprecipitate with DREAM. Additionally, co-immunoprecipitation of Sin3b and MuvB was detected in Rb/p107/p130 triple knockout mouse NIH3T3 cells, suggesting pocket proteins are dispensable for the interaction. An earlier report additionally showed an interaction of MuvB with SIN3B in all cell-cycle phases and proposed SIN3B to be an integral component of the MuvB core complex[43]. In contrast, several reports failed to detect interactions between SIN3B and DREAM/MuvB components[9,44–46].

SIN3B and its paralog SIN3A have long been implicated in the regulation of cell-cycle genes, as well as in supporting the repression and activation of a multitude of genes in other contexts[43,45–50]. Having no DNA-binding domain or enzymatic activity of their own, SIN3A and SIN3B scaffold the recruitment of chromatin modifiers like HDAC1/2, ING1/2, and KDM5A/B to histones by bridging them to transcription factors such as MAD-MAX, FOXK1, NANOG, and FAM60A[47,51]. With an amino acid identity of about 60% in humans, the two SIN3 family members have both overlapping and unique functions. Generally, loss of SIN3A results in more severe phenotypes, and cells not expressing SIN3A arrest in G2/M and enter apoptosis, which is partially caused by

activation of p53[48,52]. In contrast, Sin3b$^{-/-}$ MEFs proliferate normally but show a reduced potential to arrest under growth-limiting conditions[49]. Even though distinct SIN3A- and SIN3B-specific subcomplexes exist, both proteins have been detected at cell-cycle gene promoters in chromatin-immunoprecipitations[31,45,50,53]. Combined knockdown of SIN3A and SIN3B leads to a moderate derepression of several cell-cycle genes in differentiated C2C12 cells[45], suggesting that both proteins play a role in cell cycle-dependent gene regulation.

Here, we ask whether SIN3B and HDAC activity are generally required for cell-cycle gene repression and whether a loss of SIN3B would phenocopy disruption of DREAM repressor function as shown in LIN37-negative cells[13,14]. We find that while the requirement of LIN37 for DREAM repression persists throughout a panel of cell lines, SIN3B is uniquely tied to cell-cycle gene repression in the specific context of serum-starved T98G cells. Furthermore, while a combined loss of SIN3A and SIN3B leads to a moderate upregulation of cell-cycle gene mRNA expression in arrested HCT116 cells, this effect is independent of DREAM and E2F:RB repression. To further investigate the broader role of HDACs in cell-cycle gene repression, we treat arrested cells with the small molecule HDAC inhibitors Romidepsin and Panobinostat and find that HDACs modify histones at cell-cycle genes, but this activity does not generally impact transcription and is ultimately not essential to cell-cycle gene repression across cell lines. We conclude that DREAM and E2F:RB can repress cell-cycle genes independently of SIN3 proteins or HDAC activity.

## Results

### SIN3B is not essential for p53-dependent cell-cycle gene repression in HCT116 cells

Although the composition of the DREAM complex has been well described, only limited data on interaction partners that contribute to gene repression are available[22]. To identify proteins that are enriched at the promoters of G1/S and G2/M DREAM target genes in comparison to a group of non-cell-cycle genes, we performed an in silico association analysis using the TFEA.ChIP tool[54]. TFEA.ChIP utilizes the ReMap2022 database, which includes over 8000 quality-controlled ChIP-Seq datasets generated with more than 1200 chromatin-associated human proteins[55]. We selected a set of 109 G1/S and 132 G2/M genes (Supplementary Data 3) that we previously identified as DREAM targets[14], and we determined which chromatin-binding proteins are enriched at the promoters of these genes in comparison to a set of 4756 genes that are not DREAM targets and that are consistently expressed throughout the cell cycle.

Indicating that the analysis is robust, G1/S genes were strongly enriched for components of E2F:RB complexes (E2F1, DP1, RB), components of DREAM (E2F4, DP1, LIN9), and repressor E2Fs (E2F6, E2F7, E2F8) (Fig. 1a). The interaction of these proteins with G1/S gene promoters is expected because of the presence of the E2F element in the promoter sequence. G2/M genes, which contain the MuvB-binding CHR element, showed enrichment of DREAM proteins (E2F4, DP1, LIN9), but also of B-MYB and FOXM1, which are components of the activator MuvB complexes B-MYB-MuvB and FOXM1-MuvB (Fig. 1b). Furthermore, the CCAAT-box binding proteins NFYA and NFYB were enriched at G2/M gene promoters consistent with the observation that CCAAT-boxes are often located upstream of CHR sites[56]. Beyond these proteins, surprisingly few other chromatin-binding factors were enriched at cell-cycle gene promoters. However, several components of histone-modifying complexes like HDAC1, HDAC2, SIN3A, SIN3B, and KDM5A were significantly enriched at both G1/S and G2/M promoters in several datasets.

Considering the identification of components of SIN3:HDAC complexes at cell-cycle gene promoters[47] and that SIN3B has been connected to DREAM-repressor function[31], we aimed to address whether DREAM-dependent gene repression generally relies on recruiting SIN3:HDAC, whether the loss of DREAM function upon knockout of

## a

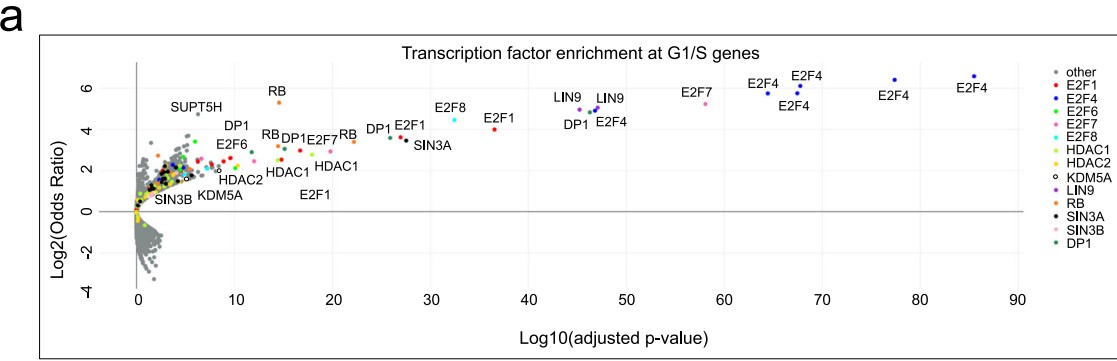

## b

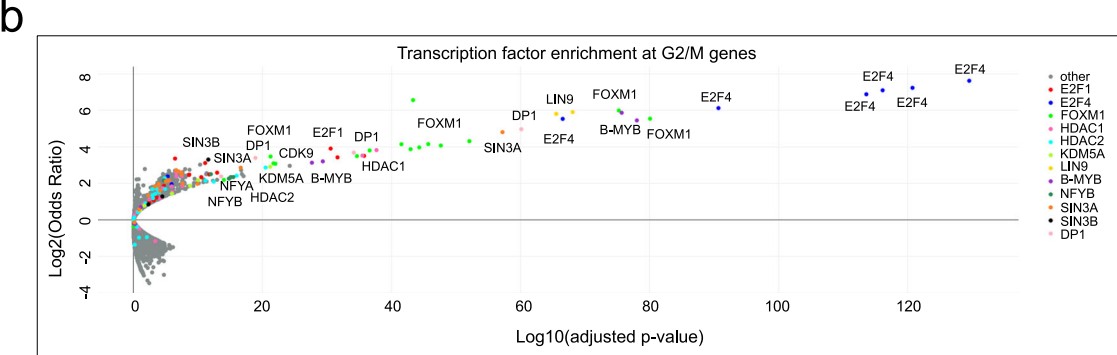

**Fig. 1 | In silico identification of chromatin-binding proteins enriched at the promoters of G1/S and G2/M DREAM target genes.** The TFEA.ChIP tool[54] was utilized for screening the ReMap2022 database for chromatin-binding proteins enriched at the promoters of (**a**) G1/S (n = 109) or (**b**) G2/M (n = 132) DREAM target genes (Supplementary Data 3). TFEA.ChIP maps ChIP-Seq peaks onto regulatory regions defined by the GeneHancer database[96] and associates these peaks with the genes regulated by those regions. The plots show Log2(Odds Ratio) versus Log10(adjusted p-value) as calculated by the TFEA.ChIP tool for each protein (represented as a single dot) in all included ChIP-Seq experiments.

LIN37 is phenocopied by loss of SIN3B, and whether LIN37/DREAM and SIN3B cooperate in repressing cell-cycle genes. To this end, we utilized wild-type HCT116 cells and several knockout lines (LIN37[-/-], RB[-/-]) we had previously generated[14] to create cells negative for SIN3B and combinations of SIN3B/LIN37 or SIN3B/RB. To minimize off-target effects, we chose a Cas9-double-nickase approach[57] and targeted regions in exon 3 or exon 4 of the *SIN3B* gene. By probing SIN3B protein expression in clonal cell lines with two independent antibodies, we confirmed the generation of SIN3B[-/-], SIN3B[-/-];LIN37[-/-], and SIN3B[-/-];RB[-/-] HCT116 cells (Fig. 2a).

Next, we used two different drug treatments that are known to stimulate cell-cycle arrest and cell-cycle gene repression. We induced DNA damage by treating wild-type and knockout lines with Doxorubicin, or we activated the p53-p21 pathway with the MDM2 inhibitor Idasanutlin. We analyzed several representative G1/S and G2/M genes and observed a strong reduction of their expression in wild-type HCT116 cells (Fig. 2b, c). Consistent with our previously published data[13,14], gene repression was impaired in LIN37[-/-] and RB[-/-] cells. Strikingly, loss of SIN3B did not have comparable effects on cell-cycle gene repression. Upon Doxorubicin treatment, repression of all analyzed cell-cycle genes either did not change significantly or was slightly stronger in SIN3B[-/-] cells. Furthermore, we did not generally observe additive effects in cells negative for LIN37 or RB together with SIN3B (Fig. 2b). Wild-type and SIN3B[-/-] cells responded equally well to the Idasanutlin treatment. We observed a slightly increased derepression of the tested cell-cycle genes in SIN3B[-/-];LIN37[-/-]cells in comparison to LIN37[-/-] cells, but no additive effects when SIN3B was depleted in RB-negative cells (Fig. 2c).

We also analyzed several G2/M and G1/S expressed proteins in untreated and Idasanutlin-treated cells by Western blot (Fig. 2d). p53-dependent repression of G2/M expressed proteins was exclusively lost in LIN37[-/-];RB[-/-] cells. Deficiency for SIN3B did not result in an upregulation of G2/M protein expression, neither when it was knocked out alone nor in combination with LIN37 or RB. The G1/S proteins CDC6 and MCM5 also did not respond to the loss of SIN3B; in contrast, MCM5 expression after Idasanutlin treatment was upregulated in all RB-negative cell lines (Fig. 2d).

Finally, we analyzed whether loss of SIN3B impairs the potential of HCT116 cells to arrest in response to Doxorubicin or Idasanutlin treatment. Both treatments induce a G1 and G2/M arrest with almost complete depletion of S phase cells in HCT116 wild-type cells (Fig. 2d, e). Loss of SIN3B did not increase S phase populations. In SIN3B[-/-];RB[-/-] cells, we observed a slight but significant increase in S phase cells relative to RB[-/-] cells (2.1% vs. 0.9%). Given these marginal effects, we conclude that loss of SIN3B does not generally impair the potential of HCT116 cells to robustly arrest in G1 and G2/M in response to p53 activation.

Since it was previously reported that SIN3B serves as an adapter protein to recruit HDAC1 to the DREAM complex in T98G cells[31], we tested whether immunoprecipitated DREAM from Idasanutlin-treated HCT116 cells contains HDAC activity. We immunoprecipitated HDAC1, SIN3B, LIN37, and RBBP4 from extracts of wild-type and knockout cells. RBBP4 is a component of MuvB as well as several chromatin-modifying complexes including the SIN3:HDAC complex[22,58]. With a luciferase-based HDACI/II-activity assay, we measured robust HDAC activity in the eluates from HDAC1, SIN3B, and RBBP4 IPs (Fig. 3a). As expected, eluates immunoprecipitated with the SIN3B antibody from extracts of SIN3B[-/-] cells showed only background activity comparable to an IgG negative control. The activity of samples immunoprecipitated with the polyclonal LIN37 antibody from wild-type extracts was slightly higher; however, HDAC activity did not change in samples precipitated from LIN37[-/-] or SIN3B[-/-] cells, which indicates that the antibody nonspecifically precipitates some HDAC activity independent of LIN37. These results were confirmed by Western blot analyses

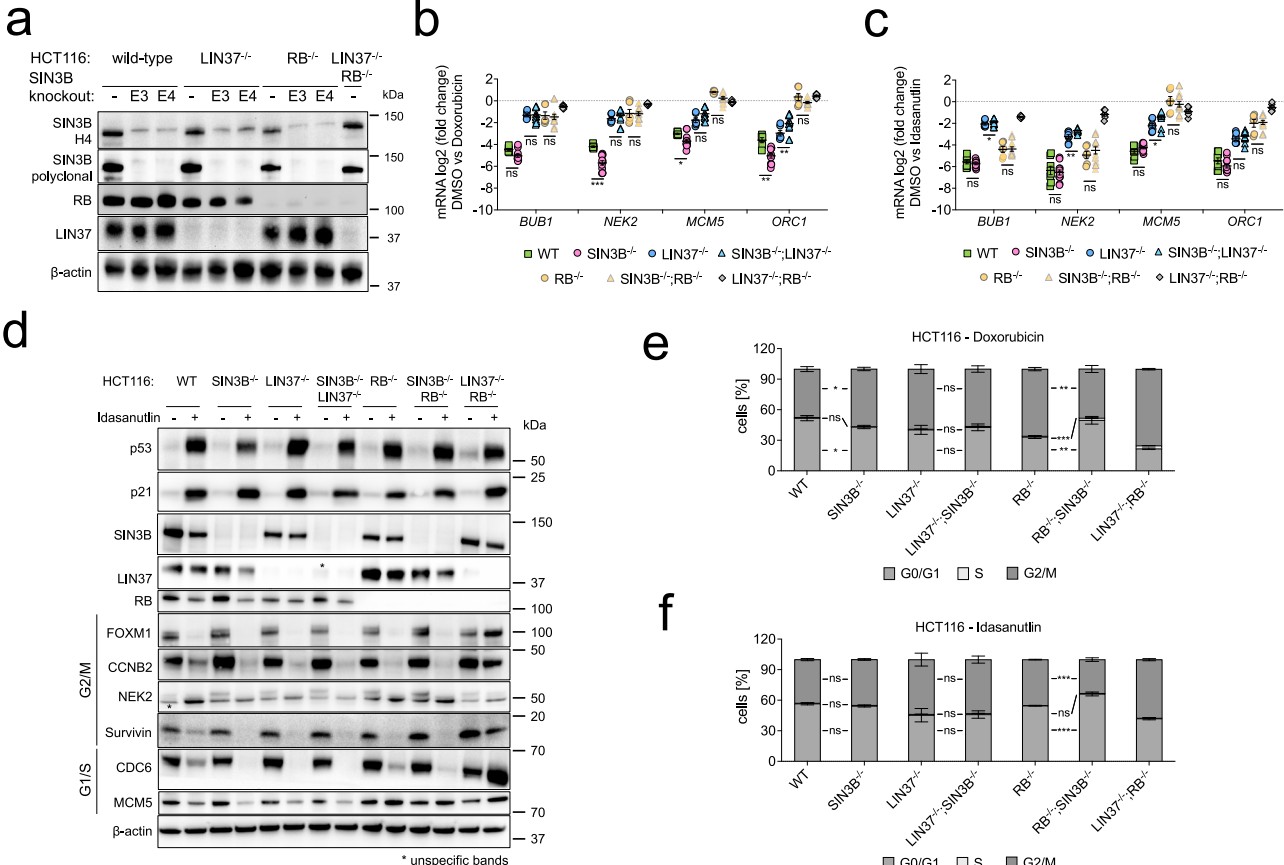

**Fig. 2 | SIN3B is not essential for the repression of G1/S and G2/M cell-cycle genes as a response to DNA damage or p53 activation in HCT116 cells. a** HCT116 cell lines negative for SIN3B were generated with a CRISPR/Cas9-nickase approach. Two pairs of guide RNAs, one targeting exon 3 and one targeting exon 4, were selected. Knockout clones were confirmed with antibodies binding epitopes within amino acids 172-228 (SIN3B-H4) or amino acids 668-758 (SIN3B polyclonal). Cells negative for SIN3B and LIN37 or RB were generated based on single knockout clones that we described earlier[14]. **b** mRNA expression of G2/M (*BUB1, NEK2*) and G1/S (*MCM5, ORC1*) cell-cycle genes was analyzed by RT-qPCR in wild-type (WT) and knockout lines after 48 hours treatment with 0.5 μM Doxorubicin. The log2 fold change between untreated and treated cells is shown. Two independent SIN3B^-/-, SIN3B^-/-;LIN37^-/-, and SIN3B^-/-;RB^-/- clones were compared with wild-type cells and one LIN37^-/-, RB^-/-, and LIN37^-/-;RB^-/- clone. The data set contains three biological

replicates, and each one was measured with two technical replicates. **c** Same experimental setup as in (**b**), but gene repression was induced by treatment with 5 μM Idasanutlin for 48 hours. **d** Protein expression of HCT116 wild-type and knockout cells after treatment with DMSO or 5 μM Idasanutlin for 48 hours was analyzed by Western blotting. One representative experiment out of three replicates is shown. **e** Cell-cycle distribution of HCT wild-type (WT) and knockout lines after 48 hours of 0.5 μM Doxorubicin treatment was analyzed by DNA staining with propidium iodide and flow cytometry. Two independent clones for each line were measured with two biological replicates. **f** Same experimental setup as in (**e**), but cells were treated with 5 μM Idasanutlin. Data in panels (**b**), (**c**), (**e**), and (**f**) are presented as mean values ± SEM. Significances were calculated with the two-tailed Student's T-Test (ns – not significant, * $p \leq .05$, ** $p \leq .01$, *** $p \leq .001$). Source data are provided as a Source Data file.

of the eluates (Fig. 3b). HDAC1 co-precipitated with SIN3B and RBBP4, but not with LIN37. In the LIN37 immunoprecipitations, we detected the MuvB component LIN9, but not HDAC1 or SIN3B. Thus, we did not observe endogenous DREAM and SIN3B/HDAC interactions in arrested HCT116 cells.

While several other publications also failed to detect an interaction between SIN3B and MuvB complex components in immunoprecipitated samples[9,44–46], binding of SIN3 and HDAC proteins to cell-cycle gene promoters has been shown in several cell lines by chromatin-immunoprecipitations (ChIP)[31,45,50,53] and is apparent in our meta-analysis of ChIP-seq data sets (Fig. 1). We wondered whether SIN3B, SIN3A, and HDAC1 binding to cell-cycle gene promoters in arrested HCT116 cells could be detected by ChIP. We performed ChIP-qPCR on samples from Idasanutlin-arrested wild-type and SIN3B^-/- cells (Fig. 3c). SIN3B was enriched at all analyzed DREAM target gene promoters in wild-type cells, and signals dropped to background level in the knockout line. Furthermore, while we detected slight changes in binding of SIN3A, HDAC1, and the DREAM component p130 to some of the six tested cell-cycle gene promoters in SIN3B^-/- cells, we did not observe any consistent trends. We conclude that the binding of

HDAC1 and SIN3A to cell cycle gene promoters does not depend on SIN3B.

Although we were not able to verify an interaction of DREAM and SIN3B in immunoprecipitation experiments (Fig. 3a, b), we considered that the binding of SIN3 proteins and HDACs to cell-cycle genes could rely on DREAM in the context of chromosomes. To deplete MuvB from the promoter of representative cell-cycle genes without introducing global perturbations in cell-cycle gene regulation, we used CRISPR/Cas9 to mutate the DREAM-binding CHR elements in the promoters of the *BUB1* and *CCNB2* genes, respectively (Fig. 3d). Loss of DREAM binding to both promoters was verified by ChIP-qPCR by analyzing LIN37 and p130. Binding of SIN3A, SIN3B, and HDAC1 to the *BUB1* promoter was reduced 2-3 fold but remained clearly above background levels. In contrast, the loss of DREAM binding to the *CCNB2* promoter led to an almost complete depletion of these proteins (Fig. 3e). mRNA expression analysis showed that inactivation of the CHRs in the *BUB1* and *CCNB2* promoters led to a highly significant loss of repression of the specific gene when the cells were arrested with Idasanutlin. In contrast, the *NEK2* gene which carries a wild-type CHR was equally strongly repressed in both cell lines, demonstrating

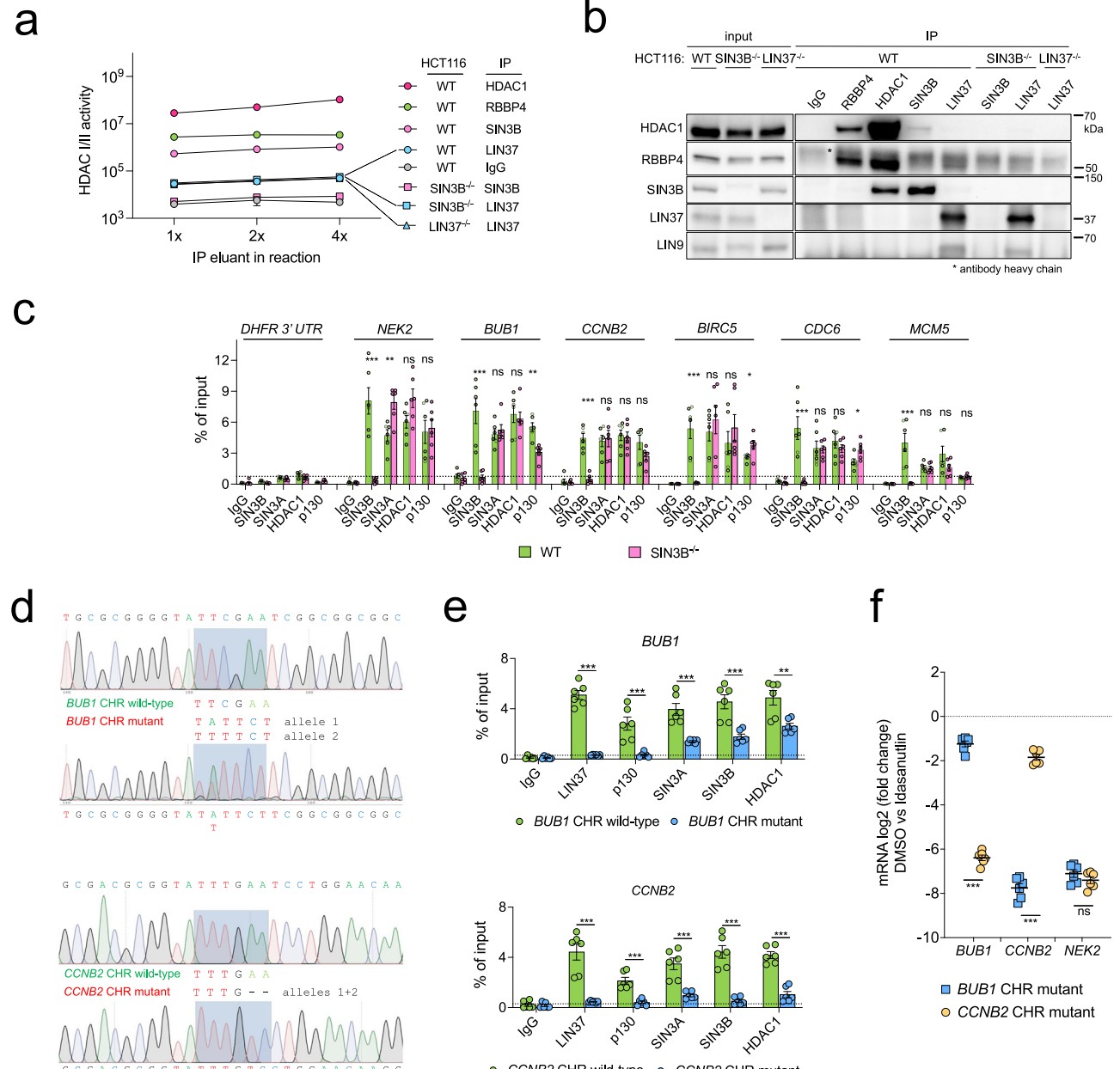

**Fig. 3 | Binding of SIN3/HDAC to DREAM and cell-cycle gene promoters.**
**a** HDACI/II activity of samples immunoprecipitated with the indicated antibodies from HCT116 wild-type and knockout cells treated with 5 μM Idasanutlin for 48 hours. Each data point contains four technical replicates (± SEM) of a representative experiment. Two biological replicates produced comparable results. **b** Protein expression and immunoprecipitation efficiency of the samples analyzed in (**a**) were evaluated by Western blotting. Two biological replicates produced comparable results. **c** ChIP-qPCR was performed to analyze the binding of SIN3B, SIN3A, HDAC1, and the DREAM component p130 to DREAM target gene promoters in wild-type and SIN3B⁻/⁻cells. A non-promoter region in the 3′ untranslated region of the *DHFR* gene (*DHFR* 3′ UTR) was amplified as a negative control. **d** HCT116 clonal cell lines containing a non-functional CHR element in the *BUB1 or CCNB2* promoter on both alleles were created by CRISPR/Cas9-mediated knock-in. Sanger

sequencing confirmed the mutation of the elements. **e** The binding of SIN3A, SIN3B, and HDAC1 to the *BUB1 or CCNB2* promoter in the cell lines described in (**d**) was studied by ChIP-qPCR. Loss of DREAM binding upon mutation of the CHR was verified by analyzing the binding of the DREAM components LIN37 and p130. Note the *BUB1* CHR wild-type cell line in this experiment is the *CCNB2* CHR mutant line and vice versa. **f** mRNA expression of *BUB1*, *CCNB2*, and *NEK2* was analyzed by RT-qPCR in the HCT116 clones carrying mutated CHR sites in the *BUB1* or *CCNB2* promoter after 48 hours treatment with 5 μM Idasanutlin. The log2 fold change between untreated and treated cells is shown. For experiments in (**c**), (**e**), and (**f**), averages of three independent experiments measured with two technical replicates are shown and presented as mean values ± SEM. Significances were calculated with the two-tailed Student's T-Test (ns – not significant, * $p \le .05$, ** $p \le .01$, *** $p \le .001$). Source data are provided as a Source Data file.

that these cells have not been altered in a way that globally affects cell-cycle gene expression (Fig. 3f). We note that our experiments show that disruption of endogenous CHR sites leads to a loss of DREAM binding and cell-cycle gene repression in response to p53 activation. We conclude that DREAM influences the recruitment of SIN3 and HDAC proteins in a promoter-dependent manner, but that

binding of these proteins to a subset of cell-cycle genes is possible without DREAM.

Taken together, these data indicate that even though SIN3B binds to the promoters of DREAM target genes, it is dispensable for cell-cycle gene repression in HCT116 cells when cell-cycle arrest is induced by activation of the p53 pathway.

## Loss of SIN3B derepresses DREAM target genes in serum-starved, but not Palbociclib-treated T98G cells

Considering that we did not observe an influence of SIN3B on the repression of DREAM target genes in HCT116 cells, we next asked whether the impaired DREAM target gene repression in SIN3B[-/-] T98G cells[31] is phenocopied by the loss of LIN37 in the same cellular system. Our CRISPR-nickase approach for generating SIN3B and LIN37 knockouts T98G cells was less efficient than in other lines, most likely because T98G is a hyperpentaploid cell line, and multiple copies of chromosome 19 that encodes for both SIN3B and LIN37 must be targeted to achieve a complete knockout. However, we were able to identify clones that did not express SIN3B, LIN37, or both proteins (Fig. 4a). Two clones of each knockout type were serum-starved, and mRNA expression of G1/S and G2/M genes was measured at 48 hours and 96 hours after serum deprivation and compared to proliferating cells (Fig. 4b). mRNA levels of all analyzed genes were strongly reduced in starved wild-type cells. In contrast, gene repression was compromised in all knockout lines. The defect in cell-cycle gene mRNA repression in serum-starved SIN3B[-/-] cells was confirmed in a time-course experiment (Supplementary Fig. 1) and is consistent with previous results[31]. The derepression of cell-cycle gene expression in cells negative for SIN3B or LIN37 was also detectable on the protein level (Fig. 4c).

To analyze cell-cycle gene repression in a setting other than serum deprivation, we treated the T98G lines with Palbociclib. We chose Palbociclib (as opposed to Idasanutlin) to directly inhibit CDK4/6 because T98G cells do not express wild-type p53. Surprisingly, while the robust loss of cell-cycle gene repression detected in starved LIN37[-/-] or SIN3B[-/-];LIN37[-/-] persisted in Palbociclib-treated cells, we did not observe a consistently significant loss of mRNA repression in SIN3B[-/-] cells after 24 or 48 hours of treatment (Fig. 4d). Additionally, Palbociclib treatment led to comparable repression of DREAM targets in wild-type and SIN3B[-/-] cells on the protein level, while increased expression was detected in LIN37[-/-] and SIN3B[-/-];LIN37[-/-] cells (Fig. 4e). We next asked whether addition of Palbociclib reinforces cell-cycle gene repression in T98G knockout lines that are serum-starved. To this end, we compared cell-cycle gene expression on mRNA and protein levels in cells that were either serum-starved for 96 hours or starved for the same period but with the addition of Palbociclib for the final 48 hours. As observed before (Fig. 4b, c), loss of SIN3B or LIN37 resulted in an elevated expression of cell-cycle genes in starved cells compared to the parental line (Fig. 4f, g). The addition of Palbociclib increased the repression in the wild-type cells, and the measured genes were repressed to the same extent or even stronger in the SIN3B knockouts. In contrast, the addition of Palbociclib to LIN37-negative serum-starved cells led only to minimal changes in cell-cycle gene expression. Palbociclib treatment for 48 hours led to a reduction of B-MYB and Survivin proteins in SIN3B[-/-] cells comparable to the wild-type cells, while protein levels remained elevated in LIN37[-/-] or SIN3B[-/-];LIN37[-/-] clones (Fig. 4f, g). Interestingly, starved SIN3B[-/-] and SIN3B[-/-];LIN37[-/-] cells showed about a 2-fold increase in S phase cells relative to wild-type and LIN37[-/-] cells (Fig. 4h) This effect could also be observed in the serum-starved and Palbociclib treated cells, which suggests that the detected S phase increase is independent of cell-cycle gene repression.

Taken together, our data demonstrate that the reduction in cell-cycle gene repression in serum-starved SIN3B[-/-] T98G cells can be bypassed by directly inhibiting CDK4/6, but only in LIN37-positive cells that can assemble a functional DREAM complex. We propose that the observed defect in serum-starved SIN3B[-/-] T98G cells is not caused by a loss of DREAM repressor function but by upstream mechanisms that result in an impaired CDK inhibition, which prevents the formation of DREAM and E2F:RB complexes.

To analyze whether endogenous DREAM contains HDAC activity in T98G cells, we immunoprecipitated HDAC1, SIN3B, and LIN37 from serum-starved T98G cells and performed HDAC activity assays with the eluates. As expected, we detected strong HDAC activity in the samples containing HDAC1 and SIN3B (Fig. 4i). The HDAC activities in eluates precipitated with the LIN37 antibody were comparable between samples obtained from LIN37 positive and negative cell lines, suggesting that the signals are nonspecific. The data obtained from HDAC assays are in line with Western blot results that show a coprecipitation of HDAC1-SIN3B and LIN37-LIN9 but no interaction of MuvB components with SIN3B or HDAC1 (Fig. 4j). As we observed in HCT116 cells (Fig. 3a, b), we did not find evidence of proteins containing HDAC activity interacting with DREAM in starved T98G cells.

## Sin3B is not essential for cell-cycle gene repression in arrested C2C12 cells

Given that we and others[31] observed defects in cell-cycle gene repression in serum-deprived SIN3B[-/-] T98G cells, we wondered whether this effect generally occurs during serum starvation. Since HCT116 cells cannot efficiently be arrested by serum deprivation and rapidly induce apoptosis following serum withdrawal[59], we created Sin3b-negative mouse C2C12 cells (Fig. 5a) and compared cell-cycle gene repression after serum starvation with wild-type and Lin37[-/-] cells (Fig. 4b)[13,14].

Starvation for 48 or 72 hours led to the repression of G1/S and G2/M genes in the wild-type cells. Loss of Sin3b did not result in defective repression of these genes. In contrast, all measured genes were significantly derepressed in serum-starved Lin37[-/-] C2C12 cells (Fig. 5b). Since it had been previously shown that Sin3b[-/-] MEFs exit the cell cycle less efficiently than wild-type cells when serum-starved[49], we analyzed cell-cycle distribution of starved wild-type and Sin3b[-/-] C2C12 cells, but we did not find significant differences (Fig. 5c). EdU incorporation assays also did not show an increase of S phase cells (Supplementary Fig. 1b). We did find a significant increase in mRNA expression in three out of four tested cell-cycle genes after Idasanutlin-treatment (Fig. 5d); however, this increase was not observed on the protein level (Fig. 5e). Based on these findings, we conclude that loss of SIN3B does not generally influence the response of cells to repress cell-cycle genes upon withdrawal of mitogenic stimuli.

## Combined loss of SIN3A and SIN3B increases the mRNA expression of multiple cell-cycle genes in arrested cells independently of DREAM or RB

Since we did not find deregulation of DREAM targets in arrested SIN3B[-/-] HCT116 cells (Fig. 2c), but these promoters still bind SIN3A (Fig. 3c), we asked whether SIN3A can compensate for the loss of SIN3B in these cells. It has been demonstrated that depletion of SIN3A results in cell-cycle arrest and apoptosis through activation of CDKN1A/p21 in a p53-dependent and -independent manner[48]. Based on these data, we refrained from knocking out SIN3A and instead chose an siRNA-based approach to reduce SIN3A expression. First, we tested the knockdown efficiency of four independent SIN3A siRNAs in proliferating HCT116 cells. All four siRNAs drastically reduced the protein expression of SIN3A, while SIN3B levels were increased (Supplementary Fig. 2a). Even though SIN3A was reduced below the detection level with all four siRNAs, accumulation of p53 and p21 only occurred with siRNAs 1, 2, and 4, suggesting that functional SIN3A remained in cells treated with siRNA 3. Cell-cycle protein expression behaved inversely to p53 and p21 levels, i.e., mitotic and S phase regulators were repressed upon transfection of SIN3A siRNAs 1, 2, and 4 (Supplementary Fig. 2a). The observed repression of cell-cycle genes was recapitulated on the mRNA level (Supplementary Fig. 2b).

Next, we analyzed whether SIN3A knockdown in Idasanutlin-treated wild-type and SIN3B[-/-] cells influenced the repression of cell-cycle genes. SIN3A protein expression was already reduced in arrested HCT116 cells without siRNA treatment, and RNA interference decreased SIN3A levels even further. Protein expression of G2/M and

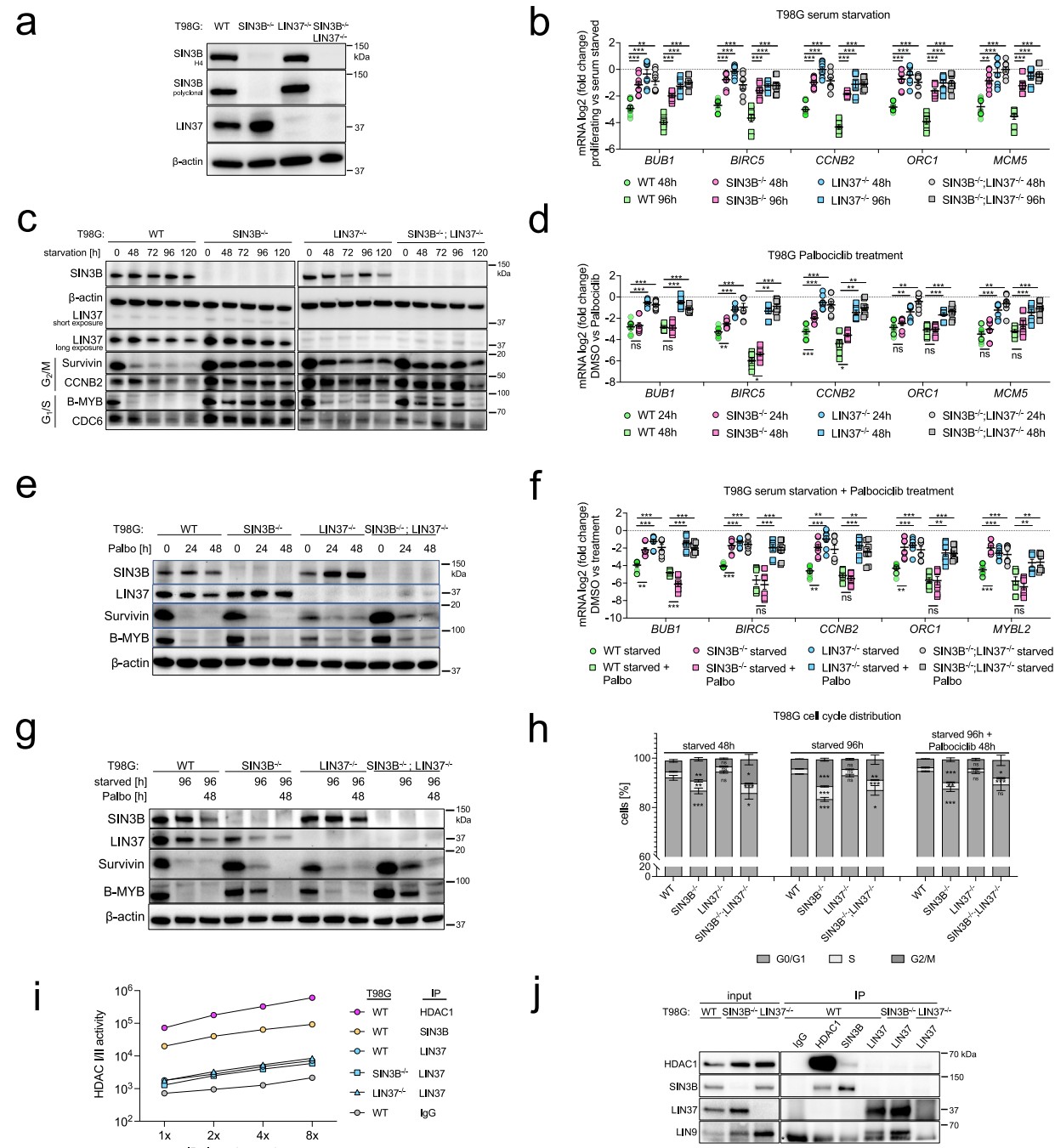

**Fig. 4 | Loss of SIN3B does not phenocopy LIN37 deficiency in T98G cells.**
**a** A CRISPR/Cas9-nickase approach to introduce mutations in exon 4 of SIN3B and exon 6 of LIN37 was applied to generate cell lines negative for SIN3B, LIN37, or both proteins. SIN3B knockout clones were confirmed with antibodies targeting amino acids 172-228 (SIN3B-H4) or amino acids 668-758 (SIN3B polyclonal). LIN37 knockout was confirmed with a polyclonal antibody raised against full-length LIN37. **b** mRNA expression of G2/M (*BUB1*, *CCNB2*, *BIRC5*) and G1/S (*MCM5*, *ORC1*) cell-cycle genes was analyzed by RT-qPCR in wild-type and knockout lines arrested by serum-starvation for 48 and 96 hours. Two independent SIN3B⁻/⁻, LIN37⁻/⁻, and SIN3B⁻/⁻;LIN37⁻/⁻ clones were compared with two wild-type (WT) clones. Averages of two biological replicates measured with two technical replicates each are given. **c** Protein expression of one of the wild-type and knockout clones measured in (**b**) was analyzed by Western blotting. Samples were derived from the same experiment and blots were processed in parallel. Similarly, **d** mRNA expression and **e** protein levels of cell-cycle genes were analyzed in two wild-type or knockout lines treated with 10 μM Palbociclib for 24 or 48 hours. **f** Indicated wild-type and knockout lines (two clones each) were serum-starved for 96 hours with or without 10 μM

Palbociclib for the final 48 hours. mRNA was measured (two biological replicates with two technical replicates each) and compared with untreated wild-type mRNA levels. **g** Samples shown in (**f**) were analyzed for protein expression by Western blotting. **h** Cell-cycle distribution of T98G wild-type (WT) and knockout lines was analyzed by DNA staining with propidium iodide and flow cytometry. Two independent clones for each line were measured with three biological replicates. **i** HDACI/II activity of samples immunoprecipitated from T98G wild-type and knockout cells serum-starved for 96 hours with the indicated antibodies. Each data point contains four technical replicates of a representative experiment. Two biological replicates produced similar results. **j** Protein expression and immunoprecipitation efficiency of the samples analyzed in (**i**) were evaluated by Western blotting. Data in Figs. (**b**), (**d**), (**f**), (**h**), and (**i**) are given as mean values ± SEM, and significances were calculated with the two-tailed Student's T-Test (ns – not significant, * $p \leq .05$, ** $p \leq .01$, *** $p \leq .001$). At least two biological replicates were performed for each Western blot experiment, and the results were similar. Source data are provided as a Source Data file.

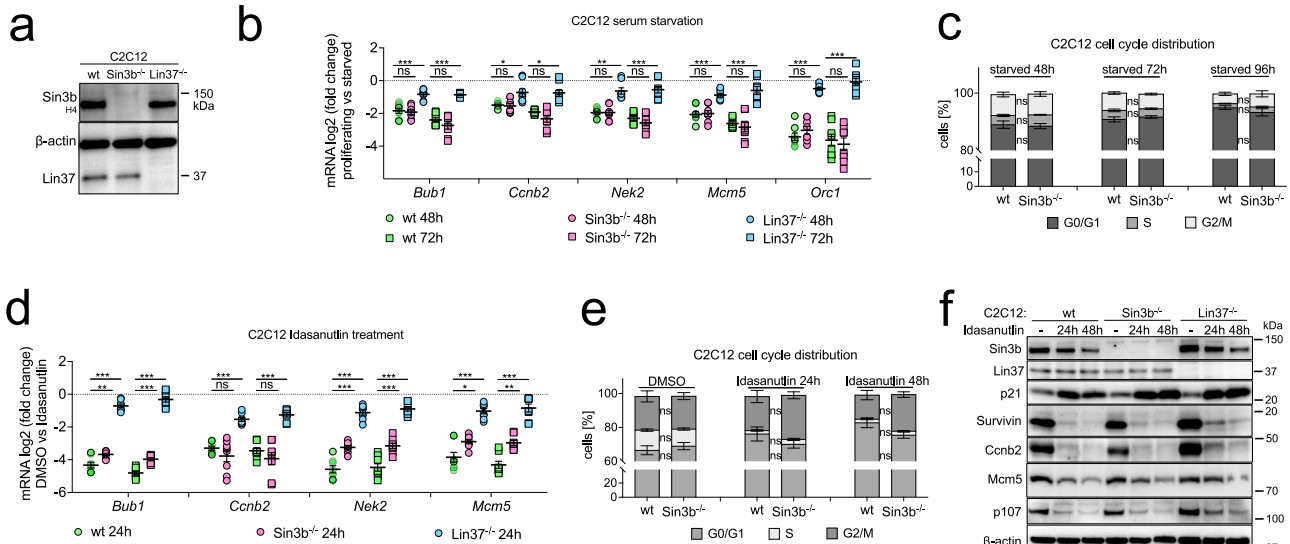

**Fig. 5 | Sin3b knockout does not phenocopy loss of Lin37 in mouse C2C12 cells.**
**a** A CRISPR/Cas9-nickase approach was applied to generate cell lines negative for Sin3b. Sin3b knockout clones were confirmed with antibodies targeting an epitope within amino acids 172-228 (SIN3B-H4). Lin37⁻/⁻ C2C12 cells were described before (Mages et al. 2017). **b** mRNA expression of cell-cycle genes was analyzed by RT-qPCR in wild-type and knockout lines arrested by serum starvation over 48 and 72 hours. Two wild-type, two Lin37⁻/⁻, and two SIN3B⁻/⁻ clones were measured with two biological and two technical replicates each. **c** Cell-cycle distribution of C2C12 wild-type (WT) and Sin3b knockout lines was analyzed by DNA staining with propidium iodide and flow cytometry. Two independent clones for each line were measured with two biological replicates. **d** mRNA expression of cell-cycle genes was analyzed by RT-qPCR in the same wild-type and knockout lines shown in (**b**) treated with 5 µM Idasanutlin for 24 and 48 hours. **e** Same experimental setup as in (**c**), but cells were arrested with Idasanutlin instead of serum-starvation. **f** The protein expression of one clone analyzed in (**d**) was studied by Western blot. Data in (**b**), (**c**), (**d**), and (**e**) are presented as mean values ± SEM. Significances were calculated with the two-tailed. Student's T-Test (ns – not significant, * $p \le .05$, ** $p \le .01$, *** $p \le .001$). Western blot experiments were performed with at least two biological replicates with similar results. Source data are provided as a Source Data file.

G1/S cell-cycle regulators was strongly repressed upon Idasanutlin treatment in both wild-type and SIN3B⁻/⁻ cells, and knockdown of SIN3A did not result in a detectable upregulation (Supplementary Fig. 2c). Knockdown of SIN3A in wild-type HCT116 cells led to minor effects regarding the mRNA expression of the analyzed cell-cycle genes, while a combined loss of SIN3B and SIN3A resulted in an upregulation throughout all analyzed genes (Supplementary Fig. 2d). Next, we performed transcriptome analyzes to identify genes deregulated in Idasanutlin-arrested WT, SIN3B⁻/⁻, SIN3A knockdown, and SIN3B⁻/⁻ + SIN3A knockdown HCT116 cells. We found comparable numbers of genes significantly (p < 0.05) up- and downregulated (≥ 1.5fold) within all three conditions when compared to wild-type cells (Fig. 6a, Supplementary Data 4). Out of 268 LIN37/DREAM target genes we had identified earlier[14], only 3 were upregulated in non-proliferating SIN3B⁻/⁻ cells, and 7 in non-proliferating SIN3A knockdown cells. However, this number increased to 57 (21%) in cells depleted of SIN3A and SIN3B (Fig. 6a, b, Supplementary Data 4). GO analyses confirmed that upregulated genes connected to cell-cycle relevant processes were only enriched in SIN3A/B-depleted cells (Fig. 6c). Interestingly, knockdown of SIN3A resulted in upregulation of gene sets connected to cilium organization and assembly, while genes upregulated in SIN3B⁻/⁻ cells only produced 7 predominantly broad terms associated with high False Discovery Rate (FDR) values. Depletion of SIN3A alone (Fig. 6d) or in combination with SIN3B (Fig. 6d) did not result in an increased population of S phase cells, indicating that the observed upregulation in cell-cycle genes is not sufficient to bypass the p53-induced cell-cycle arrest.

We then analyzed whether the observed cell-cycle gene upregulation depended on DREAM or RB. We transfected wild-type, SIN3B⁻/⁻, SIN3B⁻/⁻;LIN37⁻/⁻, and SIN3B⁻/⁻;RB⁻/⁻ lines with non-targeting or SIN3A siRNAs to analyze the expression of several cell-cycle genes that were identified as upregulated after SIN3A/B depletion in the RNA-Seq experiment by RT-qPCR. Knockouts and p53 induction were confirmed

by Western blotting (Fig. 6f). While derepression of the *CENPW* and *SGO2* genes in Idasanutlin-treated SIN3A/B-depleted cells was comparable to the effects observed in LIN37-negative cells, repression of the other analyzed genes relied more strongly on LIN37/DREAM or RB than on SIN3A/B (Fig. 6g). Furthermore, we observed that the derepression caused by loss of LIN37 or RB was generally further increased after depletion of SIN3A/B. These additive effects suggest that SIN3A/B repress cell-cycle genes independent of DREAM and RB. Comparable trends could also be observed by analyzing the expression of *BUB1*, *NEK2*, and *ORC1* (Supplementary Fig. 2d, e), which are cell-cycle genes that were not found to be significantly derepressed after SIN3A/B depletion in our RNA-seq experiment. Thus, it is likely that SIN3A/B contribute to the repression of more cell-cycle genes than we identified in the RNA-seq screen in a DREAM and RB-independent manner. The moderate upregulation of mRNA expression observed for several cell-cycle genes after the loss of SIN3A/B did not translate to detectable changes on protein level (Fig. 6f).

Taken together, loss of SIN3B or SIN3A alone did not lead to an upregulation of cell-cycle gene expression in arrested HCT116 cells. Combined depletion moderately increased cell-cycle gene mRNA expression, but this effect appeared to be independent of DREAM and RB. Moreover, effects from combined SIN3B and SIN3A depletion were minor compared to the derepression observed after loss of LIN37 or RB.

### Inhibition of HDAC activity does not broadly upregulate cell-cycle genes in arrested cells

Since we detected binding of HDAC1 to cell-cycle gene promoters in HCT116 cells independent of SIN3B (Fig. 3c), we asked whether histone tail acetylation at cell-cycle gene promoters changes during cell-cycle arrest and HDACi, and whether the repression of G2/M and G1/S genes in arrested cell lines is reduced when HDAC activity is inhibited. Since HDAC1/2 inhibition itself results in the upregulation of cell-cycle

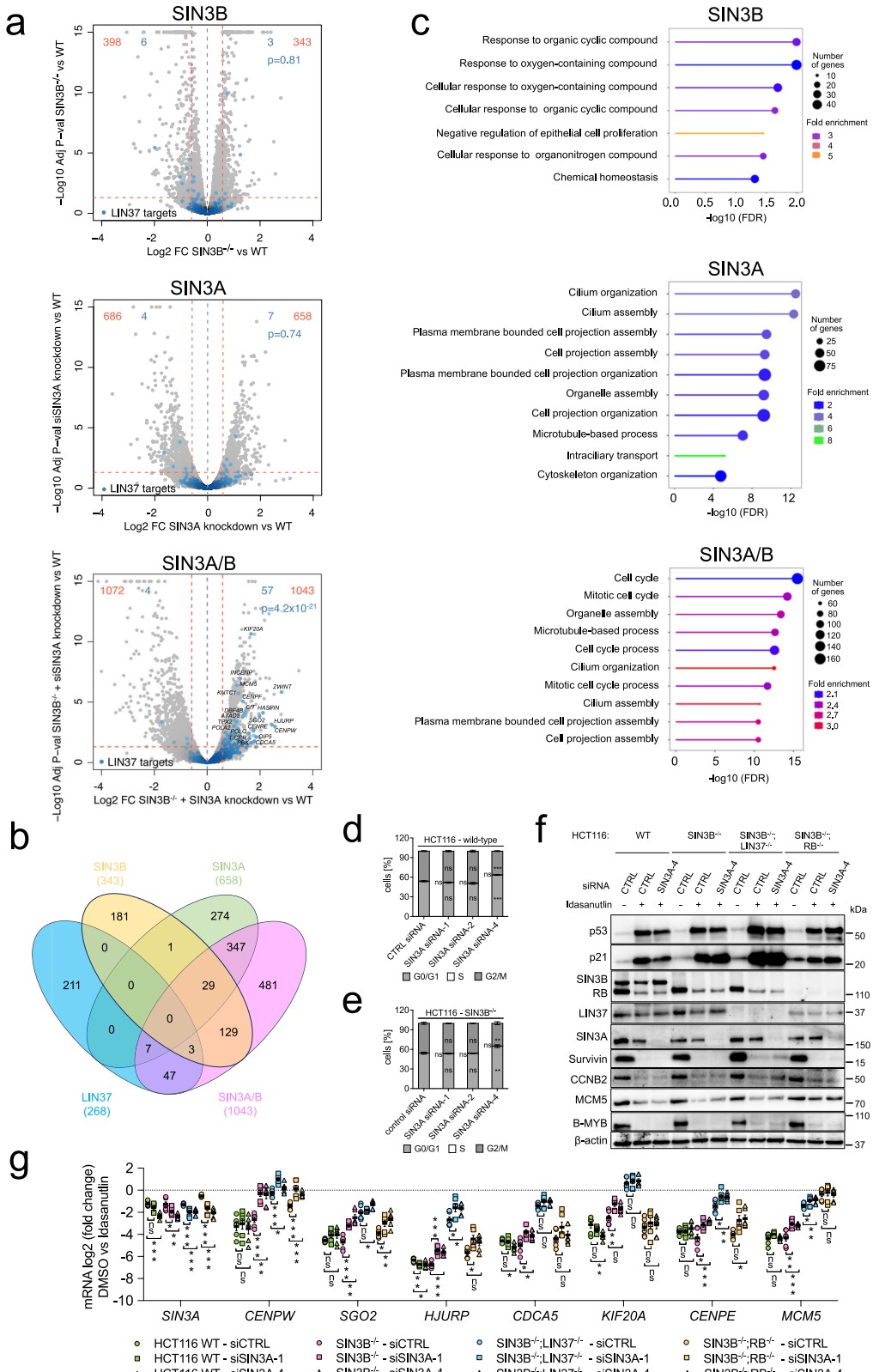

inhibitors like p21 and induces cell-cycle arrest[60–62], we arrested HCT116 cells with Idasanutlin first and then added the HDAC1/2 inhibitor Romidepsin[63,64]. As an example of a canonically activating histone mark, we analyzed whether H3K27 acetylation on cell-cycle gene promoters changes when cells were arrested with Idasanutlin and treated with Romidepsin. ChIP-qPCR analyses showed that H3K27 acetylation at the promoters of several MuvB target genes was reduced upon Idasanutlin treatment, and the addition of Romidepsin reversed this effect. (Fig. 7a). Next, we measured the expression of 19 representative G2/M and 13 G1/S genes and compared their repression in cells treated exclusively with Idasanutlin or with Idasanutlin and Romidepsin (Fig. 7b). For both groups of genes, we did not observe a significant loss of the average repression upon HDAC1/2 inhibition, although several genes like *SGO2*, *NEK2*, *E2F8*, *RBL1*, and *ORC1* were slightly, but

**Fig. 6 | Combined depletion of SIN3A and SIN3B derepresses a subset of cell-cycle genes independently of DREAM or RB.** Transcriptome analyses were performed with HCT116 wild-type (WT) and SIN3B[-/-] HCT116 cells transfected either with a non-targeting siRNA or with SIN3A siRNAs for 48 hours and treated with Idasanutlin for the final 24 hours. **a** Volcano plots show up and downregulated genes in comparison to wild-type cells. Numbers of significantly regulated genes (*p* < 0.05, fold change ≥1.5) are shown in red. Genes identified as LIN37/DREAM targets before[14] are highlighted in blue. The p-values indicate the probability that the respective overlap between LIN37 and SIN3-regulated genes could be observed by chance (hypergeometric test). **b** The number and overlap of LIN37 target genes identified in Uxa et al.[14] and genes upregulated (p < 0.05; FC ≥ 1.5) in Idasanutlin-treated HCT116 cells depleted of SIN3A, SIN3B, or both proteins. **c** GO analyzes (biological processes) of significantly upregulated (*p* < 0.05; FC ≥ 1.5) genes. The top ten hits based on their false discovery rate (FDR, as calculated by ShinyGo[95]) are shown. Cell-cycle distributions of **d** HCT116 wild-type and **e** SIN3B knockout lines were analyzed 48 hours after transfection with a control (CTRL) siRNA or three SIN3A siRNAs and Idasanutlin treatment for the final 24 hours. DNA was stained with propidium iodide and analyzed by flow cytometry. Averages of three biological replicates ± SEM are shown. **f** Wild-type, SIN3B[-/-], SIN3B[-/-];LIN37[-/-], and SIN3B[-/-];RB[-/-] knockout lines were transfected with non-targeting or SIN3A siRNAs for 48 hours, and Idasanutlin was applied for the final 24 hours. Protein levels were analyzed by Western blotting. A representative blot of cells transfected with either a non-silencing control RNA (CTRL) or SIN3A siRNA 1 is shown. A biological replicate with SIN3A siRNA 4 produced similar results. **g** mRNA levels of genes identified as significantly upregulated in SIN3A/B-depleted cells in the transcriptome analysis were evaluated by RT-qPCR. Cells were treated as described in (**f**) but additionally transfected with SIN3A siRNA 4. Averages (mean values ± SEM) of three biological and two technical replicates are given. Significances were calculated with the two-tailed Student's T-Test (ns – not significant, * p ≤.05, ** p ≤ .01, *** p ≤ .001). Source data are provided as a Source Data file.

significantly upregulated. Furthermore, except for *SGO2*, expression of genes that were upregulated in SIN3A/B-depleted cells did not increase upon HDACi. In contrast, a set of genes that had been previously reported to be upregulated in proliferating HCT116 cells after HDACi[65] showed a highly significant average increase in expression (Fig. 7b). Further demonstrating the efficacy of the drug treatments, Western blot analysis confirmed upregulation of p53 and p21 in response to Idasanutlin treatment and showed an increase of acetylated histone H3 upon Romidepsin treatment (Fig. 7c). Expression of G2/M and G1/S proteins was strongly repressed in Idasanutlin-treated cells, and addition of Romidepsin did not increase protein levels (Fig. 7c). While the observed reduction in H3K27ac levels correlates with Idasanutlin-induced gene repression, the increase that follows HDACi but not cell-cycle arrest (Fig. 7a) does not result in an upregulation of gene expression. These data indicate that cell-cycle gene repression can be maintained even when the chromatin at the promoters shows hallmarks of actively expressed genes. To analyze whether additional HDACs that Romidepsin does not inhibit influence the repression of MuvB target genes, we repeated the experiment with the pan-HDAC inhibitor Panobinostat[66] and obtained comparable results (Supplementary Fig. 3a, b). To test the effect of HDACi in additional cell lines, we treated A549 lung carcinoma cells with Idasanutlin and Romidepsin. In these cells, multiple G2/M and G1/S genes were significantly upregulated in Idasanutlin-treated cells after Romidepsin treatment, even though the average increase of expression was lower than the set of HDAC-dependent control genes (Fig. 7d). However, this upregulation of mRNA level did not lead to a detectable increase in protein expression (Fig. 7e). Treatment of Idasanutlin-arrested A549 cells with the pan-HDAC inhibitor Panobinostat led to some minor but predominantly non-significant changes in mRNA expression of G2/M and G1/S genes (Supplementary Fig. 3c), and Panobinostat-treatment also did not translate to detectable changes in protein expression (Supplementary Fig. 3d). We next analyzed effects of HDAC1/2 inhibition in arrested non-transformed mouse C2C12 cells. Comparable to HCT116 cells, mRNA expression of HDAC-dependent genes was strongly increased, while levels of G1/S and G2/M gene mRNAs were on average, not significantly upregulated (Fig. 7f). Even though several specific genes were significantly upregulated (Fig. 7f), these changes did not translate to the protein level (Fig. 7g). To analyze the effects of HDACi in a human, non-transformed cell line, we repeated the experiment in BJ-hTert cells. The addition of Romidepsin to Idasanutlin-treated cells slightly increased the expression of several G2/M genes (*CDC25C*, *NEK2*, *MKI67*), while the expression of others did not change significantly (*AURKA*, *UBE2C*, *FOXM1*) or was further reduced (*PLK1*, *CCNB2*). The set of analyzed G1/S genes generally appeared to be more strongly influenced by Romidepsin treatment, particularly *CDC6*, *ORC1*, *E2F1*, and *ZNF367*, which were significantly upregulated. However, several of the other analyzed G1/S genes did not respond to HDACi (*ATAD2*, *BRCA2*, *RAD51*, *MYBL2*), while all control genes were

significantly upregulated (Fig. 7h). Again, no change in repression of cell-cycle gene protein expression could be detected by Western blotting (Fig. 7i).

Finally, we asked whether HDACi in Palbociclib-treated T98G wild-type and knockout cells results in defects in cell-cycle gene repression since DREAM target genes can be repressed in SIN3B[-/-] cells in this context (Fig. 4). While the mRNA expression of the HDAC-dependent control gene *CTGF* was significantly upregulated in all tested clones, HDACi by Romidepsin or Panobinostat did not reduce Palbociclib-dependent repression of *BUB1* and *ORC1*. In contrast, repression of these genes was enforced in wild-type and SIN3B[-/-] cells. Considering that this effect was not observed in LIN37[-/-] cells, repression most likely occurred through upregulation of p21, CDK1/2 inhibition, and an increased formation of DREAM (Supplementary Fig. 3e).

Taken together, our results provide evidence that HDACi in arrested transformed and non-transformed cells do not generally derepress DREAM and E2F:RB target genes. We observe significant derepression of some cell-cycle genes, however the affected genes are variable between cell lines and do not affect protein levels. We conclude that HDAC involvement in cell-cycle gene regulation is not a general mechanism of repression by DREAM and E2F:RB complexes.

## Discussion

Cell-cycle-dependent gene regulation has been studied extensively for decades, but the mechanisms by which MuvB and E2F:RB complexes mediate the precisely timed repression or activation of several hundred genes remain poorly understood. In this study, we aimed to analyze to what extent HDAC activity contributes to the repression of cell-cycle genes by DREAM and E2F:RB complexes. The involvement of HDAC-containing complexes in the regulation of cell-cycle genes has been controversially discussed. The earliest reports proposing the involvement of HDACs in the repression of E2F target genes showed a direct LxCxE-dependent binding of HDAC1 to RB and suggested a dependence of RB transcriptional repressor function on HDAC1 based on promoter-reporter assays[26,28,30] or RT-PCR[29]. Interestingly, one study observed that RB-dependent repression of some of the analyzed G1/S genes relied on HDAC activity, while others were resistant[29]. In addition to HDAC1, more than 100 proteins have been reported to interact with RB[67], and a subset of them has been validated to use an LxCxE motif for binding[36]. Given that HDAC1 has a low affinity to RB in comparison to other partners that potentially compete for LxCxE-dependent RB binding[68], it remains unclear to what extent HDAC contributes to RB repressor function in particular biological settings. Furthermore, several studies provided evidence that, depending on the context, mutation of the RB LxCxE binding cleft that disrupts the multitude of potential binding interactions has only limited and, in some cases, no effects on overall cell-cycle gene repression, cell-cycle arrest or exit, and induction of carcinogenesis[37-41]. The further complexity of this system was established when the p130/p107-containing

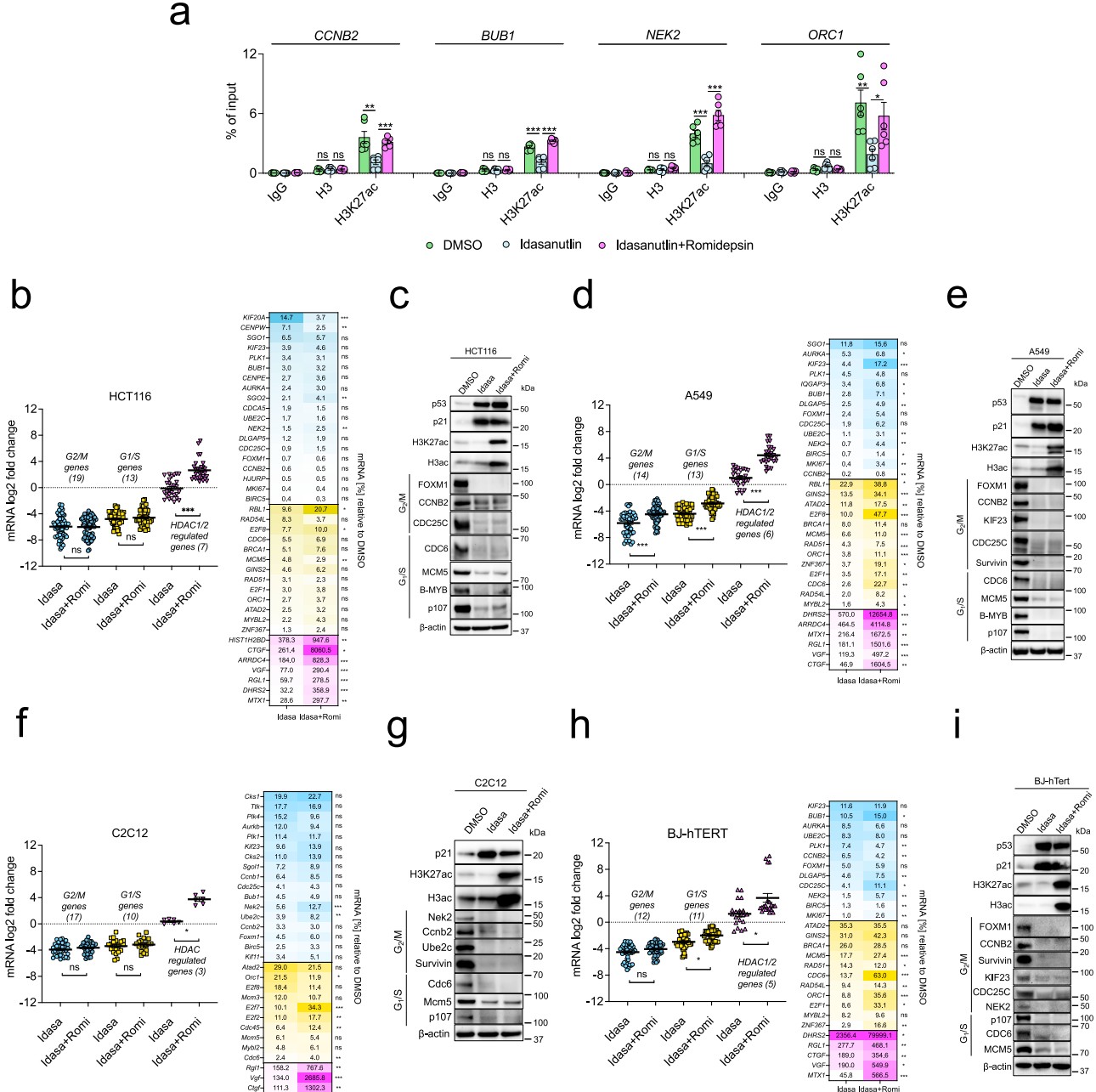

**Fig. 7 | HDAC activity is not generally required for cell-cycle gene repression in arrested cells.** In all the following experiments, cells were treated with 5 μM Idasanutlin for 48 hours and 4 nM Romidepsin for the final 24 hours. **a** H3K27 acetylation at cell-cycle gene promoters in HCT116 cells was analyzed by ChIP-qPCR. Three biological replicates with two technical replicates (mean values ± SEM) are shown. Changes in mRNA levels are presented as gene-set clusters (left) and individual genes (right) for experiments performed with **b** HCT116, **d** A549, **f** C2C12, and **h** BJ-hTERT cells. The datasets contain two biological replicates with two technical replicates each. Mean values ± SEM are shown, and significances were calculated with the two-tailed Students T-Test (ns – not significant, * $p \leq .05$, ** $p \leq .01$, *** $p \leq .001$). Protein expression and histone acetylation were evaluated via Western blotting with the indicated antibodies using (**c**) HCT116, **e** A549, **g** C2C12, and **i** BJ-hTERT cells. An additional biological replicate for each Western blot experiment produced similar results. Source data are provided as a Source Data file.

mammalian DREAM complex was discovered[9,10]. DREAM is the main transcriptional repressor of G2/M genes at CHR promoter elements, and it represses G1/S genes together with E2F:RB complexes at E2F promoter sites[69]. HDAC1 had been shown to bind p130/p107 and RB equally well and in an LxCxE-dependent manner both in vitro and when overexpressed in cells[26]. However, we later demonstrated that p130/p107 are incorporated into the DREAM repressor complex via an LxCxE-dependent interaction with LIN52[20], which makes recruitment of additional proteins through the LxCxE binding cleft unlikely. This example highlights that even though an interaction can be detected in

a specific experimental setup, it may not be relevant in every physiological context.

Given that none of the DREAM components contain any enzymatic activity and that p130/p107 cannot bind chromatin modifiers via its LxCxE binding cleft when present in DREAM, the question arises if other DREAM components can recruit such proteins. The obvious candidate for such a mechanism is the MuvB core component RBBP4, which is also found in several chromatin-modifying complexes such as PRC2, NuRD, CoRest, and SIN3[58]. However, evidence regarding binding of histone modifiers to DREAM components is limited. While IP/

MudPIT analysis after immunoprecipitating tagged p130, LIN9, LIN37, or LIN52 from T98G cells identified all DREAM components, no other interactors were detected in all samples[9]. Particularly, no SIN3B or HDAC1/2 were detected in any of the samples, even though peptides of HDAC3 were identified in LIN54 IPs and peptides of SIN3A in LIN37 IPs[9]. In contrast, tandem mass-spectrometry of samples gained by immunoprecipitating overexpressed SIN3B from immortalized MEFs identified all DREAM and MMB components except LIN52 as binding partners[31]. The strong discrepancies in the results of both studies could be caused by the expression and/or precipitation of the different proteins. However, in a recent study that extensively analyzed the interaction network of SIN3A and SIN3B, no proteins specific for MuvB complexes were found in samples precipitated with tagged SIN3A/B from HEK extracts except RBBP4[46]. This presence of RBBP4 in these SIN3A/B complexes is not surprising since RBBP4 is a known component of the canonical SIN3HDAC complex[70]. Interestingly, when Adams and colleagues did the reverse experiment and immunoprecipitated RBBP4, they readily detected SIN3A/B, but also a strong enrichment of MuvB proteins[46], which suggests that at least in this experimental setup, RBBP4 exclusively binds to either MuvB or SIN3. The conflicting results gained from IP/mass spec experiments are reflected by several studies that analyzed interactions of SIN3 and MuvB proteins by IP/Western. While we and others[9,44–46] have been unable to coprecipitate endogenous SIN3B/HDAC with DREAM complex components, two publications showed binding of SIN3B and HDAC1 to DREAM[31,43].

Here, we aimed to identify more conclusively the role of SIN3 and HDACs in DREAM and RB function and, more broadly, in the repression of cell-cycle genes. Previous genetic knockout of SIN3B increased the expression of about 100 DREAM target genes in serum-starved T98G cells, and based on the observation that H3K9 acetylation was increased at the promoters of CCNA2 and INCENP, it was suggested that SIN3B mediates the repressive function of DREAM through its ability to tether histone modifiers to chromatin[31]. In contrast, we found that loss of SIN3B does not generally derepress cell-cycle genes in HCT116 and C2C12 cells arrested by activation of the p53 pathway or serum starvation (Figs. 2, 5, and 6). We reproduced observations of a defect in cell-cycle gene repression in SIN3B[-/-] T98G cells upon serum starvation; however, CDK4/6 inhibition with Palbociclib rescued the knockout phenotype (Fig. 4), which suggests that SIN3B functions upstream of DREAM in this context. In contrast, the loss of DREAM repressor function we found previously in arrested LIN37[-/-] NIH3T3, HCT116, and C2C12 cells[13,14] persisted in serum-starved and Palbociclib-treated LIN37[-/-] T98G cells (Fig. 4). Despite these diverging results regarding functional interactions between DREAM and SIN3:HDAC complexes, consistent data have been published showing that SIN3A, SIN3B, and HDAC1/2 bind to G1/S and G2/M cell-cycle gene promoters[31,45,50,53]. These data are in line with our in silico analysis (Fig. 1) and ChIP-qPCR results from arrested HCT116 cells (Fig. 3).

It remains to be elucidated how SIN3 proteins get recruited to cell-cycle gene promoters and how they contribute to repression, given that they neither contain a DNA-binding domain nor enzymatic activity[51]. Our finding that SIN3 and HDAC protein binding to the BUB1 promoter is reduced, but still clearly detectable, in the absence of DREAM (Fig. 3e) supports a model in which recruitment of SIN3 and HDAC can be independent of but perhaps indirectly stimulated by DREAM. This data is in line with a study that showed that binding of SIN3B to several cell-cycle gene promoters persisted in p130[-/-];p107[-/-] and Rb[-/-] serum-starved MEFs (Rayman et al., 2002). Furthermore, in differentiated C2C12 cells, tiling array data showed that E2F4 binding to promoters is centered around the TSS, while peaks of SIN3A and SIN3B are shifted about 200 bp downstream of the TSS[45]. Based on these results, it appears unlikely that SIN3 proteins get recruited to cell-cycle genes via direct binding to DREAM or RB.

Our data generated with HCT116 cells depleted of SIN3B and SIN3A show that these proteins contribute to the repression of a subset of cell-cycle genes and can substitute for each other (Fig. 6). These results are supported by the findings that combined knockdown of SIN3A and SIN3B in differentiated C2C12 cells resulted in about a 3-fold upregulation of several cell-cycle genes[45], which reflects the level of derepression we found in SIN3A/B-depleted arrested HCT116 cells (Fig. 6). Since we found an additional increase of derepression in arrested LIN37[-/-] and RB[-/-] cells depleted of both SIN3 proteins (Fig. 6g, Supplementary Fig. 1E), our results suggest that even though DREAM supports binding of SIN3 proteins to some cell-cycle gene promoters (Fig. 3e), their repressor function is independent of DREAM and RB. We suggest a model where DREAM binding modulates chromatin structure in a way that allows efficient binding of co-repressor complexes. Since HDACi did not phenocopy loss of SIN3A/B (Figs. 5, 6, Supplementary Figs. 1, 2), it is also likely that the observed SIN3-dependent effects do not rely on recruitment of HDACs, but on other chromatin-modifying enzymes or transcriptional cofactors that have been described as interactors of SIN3 proteins[51].

Many studies have analyzed the effects of HDACi on transformed and non-transformed proliferating cells, and generally, treatment with HDAC inhibitors increases the expression of anti-proliferative and pro-apoptotic genes, represses the expression of pro-proliferative genes and results in cell-cycle arrest and apoptosis[62,71–79]. These results can be recapitulated by combined knockout of HDAC1 and HDAC2[61]. Since cancer cells are generally more sensitive to HDACi than non-transformed cells, HDAC inhibitors are promising drugs in cancer therapy. Four molecules have been approved by the FDA, and there are a multitude of ongoing clinical trials[7]. A central mechanism that initiates cell-cycle arrest and apoptosis following the loss of HDAC1/2 activity is an increased expression of the CDK inhibitors p21, p27, and p57[61,80]. p21 expression is further stimulated through acetylation and stabilization of p53[7]. Reduced CDK activity from high inhibitor levels results in the accumulation of unphosphorylated pocket proteins followed by the formation of E2F:RB and DREAM complexes. We and others have shown that these complexes are essential for inducing G0/G1 arrest[11,13,14,81,82]. Therefore, if HDAC activity is necessary for the repression of cell-cycle genes by RB and DREAM, as several studies have shown[23–31], how could it be that HDACi results in the arrest of cancer cells? A possible explanation of this paradox could be that E2F:RB and DREAM can partially repress their target genes without recruiting HDAC activity, but HDACs are essential to completely shut down cell-cycle gene transcription. In this context, gene expression data originating from proliferating cells treated with HDAC inhibitors are not particularly helpful since they do not provide information on whether genes are completely repressed upon treatment. Thus, we addressed this question by inducing strong repression of cell-cycle genes through activation of the p53-p21 pathway and then added HDAC inhibitors to analyze if repression is relieved. Even though we found that HDACi increases histone tail acetylation at cell-cycle gene promoters (Fig. 7a), we did not detect a global derepression in sets of representative G1/S and G2/M genes (Fig. 7, Supplementary Fig. 3).

Our results are another example that suggests that histone marks previously associated with actively expressed genes do not directly cause or even always correlate with transcriptional activation[3]. Many of the original data connecting pocket-protein-dependent repression to HDAC activity were generated with in vitro approaches, reporter assays, and over-expressed proteins[23–31]. Using such artificial methods may have led to an overinterpretation of the results, and slight changes found in the mRNA expression of a few tested genes were later generalized to the regulation of cell-cycle genes. In contrast, our data suggest that HDAC activity is not generally required for cell-cycle gene repression during reversible cell-cycle arrest and that alternative mechanisms like RB-dependent inhibition of activator E2Fs[33,34], stabilizing nucleosomes by DREAM[19], and potentially recruitment of other chromatin-modifying or nucleosome-remodeling proteins are

sufficient to induce robust repression. However, HDAC activity may be involved in fine-tuning the activity of subsets of cell-cycle genes under specific physiological conditions[32], and by influencing chromatin structure, DREAM may play an indirect role in HDAC recruitment. Furthermore, in contrast to reversible cell-cycle arrest, HDAC activity may be required for the permanent silencing of cell-cycle genes during terminal cell-cycle exit. This topic has yet to be well-studied, but several transcriptome data sets derived from terminally differentiated cells negative for HDAC1/2 or treated with HDAC inhibitors are available and suggest that HDAC activity is also not generally required in this context[83,84]. Taken together, it remains to be elucidated in which biological contexts SIN3 proteins and HDAC activity substantially contribute to cell-cycle gene regulation.

## Methods

### Cell culture and drug treatment

Cell lines were obtained from ATCC. Authentication was performed by ATCC and according to ATCC verification procedures, which include Mycoplasma detection, STR profiling, and Sanger sequencing. HCT116 (#CCL-247), T98G (#CRL-1690), C2C12 (#CRL-1772), A549 (#CCL-185), and BJ-hTert (#CRL-4001) wild-type and knockout cells were grown in Dulbecco's modified Eagle's medium (Gibco, #10569044) supplemented with 10 % fetal calf serum (Corning, #MT35010CV) and penicillin/streptomycin (Gibco, #15140122). Cells were maintained at 37 °C and 10% $CO_2$ and were tested negative for mycoplasma contamination by PCR with a mixture of primers that have been described previously[85]. For induction of p53, cells were treated with Idasanutlin (5 μM; R&D Systems, #12-35-22-07) or Doxorubicin (0.5 μM; Selleckchem, #E2516). T98G cells were starved in DMEM containing 0% FBS, and C2C12 cells were starved with DMEM containing 0.1% FBS. CDK4/6 inhibition was performed with Palbociclib (10 μM, Selleckchem, #S4482). Histone deacetylases were inhibited with Romidepsin (4 nM; Active Motif, # 14083) or Panobinostat (20 nM; Selleckchem, # S1030).

### Generation of knockout cell lines by CRISPR/Cas9 nickase

$SIN3B^{-/-}$, $SIN3B^{-/-};LIN37^{-/-}$, $SIN3B^{-/-};RB^{-/-}$ HCT116 cells, $SIN3B^{-/-}$, $LIN37^{-/-}$, $SIN3B^{-/-};LIN37^{-/-}$ T98G cells, and $SIN3B^{-/-}$ C2C12 cells were created by CRISPR/Cas9 nickase, applying the pX335-U6-Chimeric_BB-CBh-hSpCas9n(D10A) vector[57,86]. Mutations were introduced into exons 3 or 4 of the human SIN3B gene, exon 5 of the mouse Sin3b gene, and exon 6 of the LIN37 gene. Sequences of the oligonucleotides are provided in Supplementary Data 1. Cells were transfected with px335 vectors together with an EGFP-expressing plasmid for selection of the transfected cells using PEI 25 K (Polysciences Inc., #NC1014320). 48 hours post-transfection, EGFP-positive cells were isolated with a FACSAria Fusion Flow Cytometer (BD Biosciences). Cells were incubated in 6-well plates for 24 hours and afterward transferred to 96 cell plates after serial dilution to obtain clonal cell lines. Knockout clones were identified by SDS-PAGE and Western blot.

### Generation of knockin cell lines by CRISPR/Cas9 and ssDNA

HCT116 cells with mutant CHR elements in the *BUB1* or *CCNB2* promoters were created by transfection with the pX330-U6-Chimeric_BB-CBh-hSpCas9 vector[57,86] together with ssDNA homology-directed repair (HDR) donor templates and an EGFP-expressing plasmid using PEI 25 K (Polysciences Inc., #NC1014320). Alt-R HDR Donor Oligos were designed with the Alt-R™ HDR Design Tool and ordered from IDT. Transfected cells were isolated and cultivated as described above. Mutation of CHR elements on both alleles was confirmed by PCR amplification and Sanger Sequencing.

### RNA interference

HCT116 cells were cultivated in 6-well plates and transfected with 20 nM SIN3A siRNAs (Horizon Discovery, #MQ-012990-00-0002) or a non-targeting control siRNA (siGENOME Non-Targeting siRNA 5, Horizon Discovery, # D-001210-05-05) and 5 μl Lipofectamine RNAiMAX (Invitrogen, #13778075) in a total volume of 2 ml in antibiotics free 10% DMEM. 24 hours after transfection, cells were treated with either DMSO or 5 μM Idasanutlin (R&D Systems, #12-35-22-07) for an additional 24 hours before cells were harvested for protein and RNA extraction as well as flow cytometry analysis.

### RNA extraction, reverse transcription, and semi-quantitative real-time PCR

Total RNA was isolated with the Direct-zol RNA MiniPrep Kit (Zymo Research, #R2053). One-step RT-qPCR was performed using the GoTaq 1-Step RT-qPCR System (Promega, #A6020) and MicroAmp Fast Optical 96 Well Reaction Plates (Applied Biosystems, #4346907) on a Quantstudio 3 Real-Time PCR cycler (ThermoFisher Scientific). See Supplementary Data 1 for primer sequences.

### SDS-PAGE and Western blot

Whole-cell extracts prepared with RIPA buffer (10 mM Tris-HCl pH 8.0, 150 mM NaCl, 1 mM EDTA, 0.1% SDS, 0.1% Sodium Deoxycholate, 0.1% TritonX) or immunoprecipitated samples were analyzed by SDS-PAGE and Western blot following standard protocols[87]. See Supplementary Data 2 for the list of antibodies used for Western blotting.

### Chromatin immunoprecipitation (ChIP)

Cells were harvested, cross-linked with PBS (Gibco, # 14190250) supplemented with 1% paraformaldehyde (Electron Microscopy Sciences, #50-980-487), and quenched with 125 mM Glycine (Fisher Scientific, # BP3815). Nuclei were isolated using Buffer A (Cell Signaling, #7006 S) and Buffer B (Cell Signaling. #7007 S). MNase enzyme was prepared in-house using Addgene plasmid # 136291[88]. Nuclei were MNase-treated on ice for 30 minutes, followed by 15 minutes of incubation at 37 °C and 5x direct sonication for 1 s to create ~300 bp chromatin fragments. Protein-DNA complexes were immunoprecipitated with the indicated antibodies overnight at 4 °C and bound to Pierce protein A/G magnetic beads (ThermoScientific, #PI88803). Beads were subsequently washed with the following buffer types in order: 6x RIPA (10 mM Tris-HCl pH 8.0, 1 mM EDTA, 0.1% SDS, 0.1% Sodium Deoxycholate, 0.1%TritonX) supplemented with 140 mM NaCl, 3x RIPA supplemented with 500 mM NaCl, 3x LiCl buffer (10 mM Tris-HCl pH 8.0, 250 mM LiCl, 1 mM EDTA, 0.5% Sodium Deoxycholate, 0.5% Triton X), and 3×10 mM Tris-HCl 8.0 (salt-free). Precipitants were eluted twice with 150 μl elution buffer (10 mM Tris-HCl 8.0, 5 mM EDTA, 300 mM NaCl, 0.6% SDS) with 30 s vortexing and 15 min incubation at 37 °C. Eluants were treated with RNaseA (Thermo Scientific, #FEREN0531) for 30 minutes at 37 °C, then treated with Proteinase K (Thermo Scientific, #EO0491) for 1 hour at 55 °C, and reverse cross-linked at 65 °C overnight. DNA was purified using Zymo DNA Clean & Concentrator-5 kits (Zymo Research, #77001-152). qPCR was performed with the GoTaq qPCR Master Mix (Promega, #A6001) on a Quantstudio 3 Real-Time PCR System (ThermoFisher Scientific). See Supplementary Data 1 for primer sequences and Supplementary Data 2 for antibodies.

### Flow cytometry

The DNA content of HCT116, T98G, and C2C12 cells was analyzed by staining with propidium iodide (PI). Cells were fixed in 70% ethanol overnight, pelleted at 1000x g for 5 mins, washed once with PBS, and resuspended in 60–100 μl of FX cycle PI/RNase stain solution (Invitrogen, # F10797) before being analyzed on the flow cytometer. EdU assays were performed with the Click-iT® EdU Alexa Fluor 647 Flow Cytometry Assay Kit (ThermoFisher Scientific, #C10420) according to the manufacturer's instructions. Flow cytometry analysis was performed on a Beckman Coulter CytoFLEX and data were analyzed using Flow Jo software (BD). Examples for gating/sorting strategies are provided in Supplementary Fig. 4.

## HDAC activity assay

Cells were lysed in lysis buffer (50 mM Tris pH 8.0, 10 mM MgCl2, 0.2% Triton X, 300 mM NaCl) by 5x direct sonication for 1 s. Lysates were clarified by centrifugation (13,000 rpm, 10 min, 4 °C), and NaCl concentration was adjusted to 150 mM. 2–3 μg of antibodies were incubated with Pierce protein A/G magnetic beads (ThermoScientific, #PI88803) for 1 hour on a rotator at 4 °C. Beads were washed 3x with the buffer previously described with 150 mM NaCl. 5 mg protein extracts were incubated with the antibody-bound beads for 3 hours on a rotator at 4 °C. Beads were washed 5x with lysis buffer containing 150 mM NaCl and resuspended in 500 μl HDAC-Glo buffer (Promega, #G648A). Beads were serially diluted with HDAC-Glo buffer, and 20 μl of each dilution was transferred into a 384-well, white, flat-bottom plate. HDAC activity of three technical replicates was measured using the HDAC-Glo I/II Assay (Promega, #G6420) following the manufacturer's protocol on-bead with an Envision plate reader (Perkin Elmer). The remaining beads were boiled in Laemmli buffer and subjected to SDS-PAGE and Western blot analysis.

## Next generation sequencing and transcriptome analysis

Total RNA was isolated from HCT116 cells with the Direct-zol RNA MiniPrep Kit (Zymo Research, #R2053). Library preparation, rRNA depletion, and Illumina sequencing were performed at Genewiz/Azenta Life Sciences. Reads were trimmed with trimgalore (version 0.6.10, https://www.bioinformatics.babraham.ac.uk/projects/trim_galore/) using cutadapt version 4.2[89] and fastqc v0.12.1 (http://www.bioinformatics.babraham.ac.uk/projects/fastqc) for quality control. GNU parallel was used for parallelization[90]. After trimming, 0.5 % to 0.7% of reads were too short to be considered mappable (< 20 nt). Trimmed reads were mapped to the hg38 genome using segemehl (version 0.3.4)[91] using standard parameters and the -S option to be able to map spliced reads. 42 to 53 million reads were mapped per sample (93 % to 96 % of all reads). Between 86 % and 91 % of all reads were mapped uniquely. The mapped reads were annotated using featureCounts[92] version 2.0.3 against the gencode v.27 annotation, using the following parameters: -p -t exon -g gene_id. The resulting reads per gene counts were normalized and analyzed using DESeq2[93] to find differentially expressed genes. Comparisons of conditions and normalizations were done pairwise. All reported and plotted p-values are multiple hypotheses adjusted by DESeq2 with the Benjamini and Hochberg method. To calculate enrichment of LIN37 targets among up-regulated genes (Fig. 6a) we used the hypergeometric test as implemented in the phyper function in the stats package in R[94]. GO term analyses were performed with ShinyGo[95], and Venn diagrams were built with Venny (https://bioinfogp.cnb.csic.es/tools/venny/index.html).

## Statistics and reproducibility

Numbers of sample sizes and replicates, as well as information regarding statistical tests, are provided in the figure legends. Generally, experiments were repeated with biological samples that were collected in triplicate or duplicates with more than one clone. RNA-seq analyses were performed with three biological replicates for wild-type, SIN3B⁻/⁻, SIN3A knockdown, and SIN3B⁻/⁻;SIN3A knockdown cells. One of the wild-type samples had to be excluded from the RNA-seq experiment because it had a very high adapter content. The final mappable reads of that sample were only a fraction of the other samples. PCA analysis revealed that this sample is a severe outlier, accounting for over 50% of the variance between 12 replicates and 4 conditions. We therefore excluded that wild-type sample from the further analysis. The experiments were not randomized. The investigators were not blinded to allocation during experiments and outcome assessment.

## Reporting summary

Further information on research design is available in the Nature Portfolio Reporting Summary linked to this article.

## Data availability

RNA-Seq data generated for this publication were deposited at the Gene Expression Omnibus under the accession number GSE240734. Source data are provided in this paper.

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

## Acknowledgements

Technical support was provided by Beverley Rabbitts, UCSC Chemical Screening Center, RRID SCR_021114, and Bari Holm Nazario, IBSC Shared Stem Cell and FACS Core, RRID SCR_021149. Acquisition of the Cyto-FLEX Flow Cytometer was supported by the NIH S10 Instrumentation Grant Program (award number S10OD030423). This work was supported by grants from the National Institutes of Health to S.M.R. (R01GM124148 and R35GM145255). A.K.B. is supported by the Tobacco-Related Disease Research Program (27DT-0005C). T.U.W. is supported by a Ruth L. Kirschstein Predoctoral Fellowship from the National Cancer Institute (F31CA254090). K.M.R. is supported by NIH award K12GM139185 and the UCSC Institute for the Biology of Stem Cells (IBSC). Q.N.L. is supported by the Maximizing Access to Research Careers training program award T34GM140956.

## Author contributions

G.A.M. and S.M.R. conceived and supervised the study. A.R. evaluated the RNA-seq data. A.B., M.R.S, G.A.M., K.M.R, Q.N.L., T.U.W., C.E.M., and M.W.M. acquired all other data. G.A.M., S.M.R., A.B., and A.R. wrote the original draft of the manuscript. All other authors reviewed and edited the manuscript.

## Competing interests

The authors declare no competing interests.
