## [Peer Review File · Nature Communications]

HDAC activity is dispensable for repression of cell-cycle genes by DREAM and E2F:RB complexesReviewers' Comments:

Reviewer #1:

Remarks to the Author:

Barrett et al describe a very thorough investigation of the roles played by Sin3B and Sin3A proteins, and by HDAC activity, in the regulation of cell cycle gene expression in human cell lines.

This is an important study. As the authors describe in their introduction, and in the discussion, many published studies have proposed that HDAC activities are recruited by RB and/or DREAM complexes to the promoters of cell-cycle regulated promoters, and that these proteins are important for the cyclic patterns of gene expression. But different studies have reported different results, and from reading the literature it is impossible to say with confidence exactly what the true roles of these proteins are.

Barrett et al have revisited this subject using gene targeting technology, and they are able to provide some clear answers. To some readers the results may seem disappointing because many of the results are negative. Barrett et al show that the genetic knockout of SIN3B did not cause the derepression of cell-cycle genes in non-proliferating HCT116 and C2C12 cells. The combined loss of SIN3A and SIN3B resulted in only a moderate upregulation of expression from several cell-cycle genes in arrested HCT116 cells. Supporting this Barrett et al show that HDAC inhibition does not induce a general upregulation of RB and DREAM target gene expression in arrested transformed or non-transformed cells. The results are especially convincing because Barrett et al are able to contrast these findings with the results that they detect in Lin37 deficient cells. Using compound mutants, they are able to make a good case that the lack of effects that they see with SIN3A/SIN3B/HDAC cannot be easily explained by redundancy. There are many interesting details but the main conclusion that I took from this work is that the claims that many studies have made about the roles of SIN3A, SIN3B and HDAC proteins at cell cycle genes have been overstated. It is very important that this is clarified in the literature. In contrast, the model that the authors recently proposed, that DREAM complexes position or stabilize nucleosomes over the transcription start site, seems more likely to be true. This model may apply to repression by both DREAM- and RB-complexes, but presently it is not clear (at least, not to me) how the authors will prove that this is the true mechanism of action.

I had just a couple of questions and a comment:

1. For the in silico analysis shown in Figure 1, what is the size of the promoter fragments that have been examined? This is an important detail and should be mentioned in the Figure legend. It would be interesting to know how the enrichment scores change if the analysis centers only on the DREAM or RB binding sites. Having seen the functional data, one wonders if the binding profile for the components of HDAC complexes can be separated in silico from the DREAM and E2F/RB proteins.

2. Cell cycle data is really important for these experiments. Because the authors are looking at cell cycle dependent transcription any changes in cell cycle position, even small changes, could alter patterns of expression. Perhaps the controls are in here and I just haven't searched carefully enough, but I couldn't find the data showing the knockouts had no effect on cell cycle distribution, and that the drug treatments had exactly the same effects (on DNA damage and cell cycle progression) that they have in the control cells. I assume that there are no major changes (because the main observation is the lack of changes in gene expression) but these controls need to be included.

3. In the discussion please can the authors comment on whether they think that the conclusions that they draw extend to all cell types, and all physiological contexts, and whether they think that DREAM, MMB, and RB/E2F complexes have a single mechanism of action. There are hints in the text, but it might be helpful if the authors were to address these points directly.

Reviewer #2:

Remarks to the Author:

Noteworthy results

o Barrett et al. show that SIN3B is not required for the inhibition of some cell cycle genes induced by DNA damage or P53 activation.

o Barrett et al. show that SIN3B is required for some cell cycle gene repression when grown in serum starved conditions. Inducing DNA damage in LIN37 KO also leads to derepression of cell cycle genes. They also show that LIN37 does CoIP with a complex containing HDAC activity in serum starved conditions.

o Barrett et al. show that LIN37 KO in mouse cell line C2C12 does not phenocopy SIN3B KO at the mRNA level but not the protein level, in response to serum starvation.

o Barrett et al. show that SIN3A function is partially redundant with SIN3B function independent of RB or LIN37.

o Barrett et al. show that Class I HDACs are not involved in the repressive mechanism of LIN37 on cell cycle genes.

The work is important and will advance the scientific knowledge of the mechanism in which the DREAM complex regulates cell cycle genes.

Overall, the work somewhat supports the conclusions drawn. I have noted below where the data did not support the conclusion drawn and where additional experiments should be done:

Overall note: Three biological replicates should be done when appropriate. Average of all biological replicates should be presented when representing data as a graph.

- "Gene repression was impaired in LIN37^{-/-} and RB^{-/-} cells, and we detected a stronger loss of repression for G2/M genes in LIN37^{-/-} cells and for G1/S genes in RB^{-/-} cells" Cannot make this statement as statistical significance between LIN37 KO / RB KO cells and WT cells was not done.
- "Importantly, combined loss of SIN3B and RB did not reflect the almost complete loss of repression observed in LIN37^{-/-};RB^{-/-} cells" Cannot make this statement as statistical significance was not done. In any case, gene expression of MCM5 and ORC1 in the cells with dual SIN3B and RB KO is not very different from LIN37 RB dual KO cells.
- Since the authors are making conclusion by western blot analysis, quantification of bands should be done for Fig. 2D
- Fig. 2E. The X and Y axis are relative to what? This information should be added to the legend
- "mRNA levels of all analyzed genes were strongly reduced in starved wild-type cells" The authors need to be clear that the statistical significance was only measured at 120 hrs, thus this statement is misleading. Alternatively, you can measure the statistical significance of all timepoints. This is important because later experiments use different starvation times. Additionally, the *** indicating significance is very small and cannot be read.
- Fig 3B. The authors should add a sentence on what mRNA % means, as fold change is the conventional way to represent qPCR data.
- Fig. 3C. The WB is confusing and not consistent with the rest of the paper. The loading control should be at the bottom.
- Fig. 3D panel 1. Y axis is different from the rest of the panel.
- "Additionally, Palbociclib treatment led to comparable repression of DREAM targets in wild-type and SIN3B^{-/-} cells on the protein level, while increased expression was detected in LIN37^{-/-} and SIN3B^{-/-};LIN37^{-/-} cells" This statement is not true as the amount of Survinin and B-MYB at 0 hr is different from WT and SIN3B KO, therefore the change is due to the KO and not treatment. Additionally, the authors needs to quantify western blots when using western blots to quantify expression levels.
- "In contrast, addition of Palbociclib to LIN37-negative serum-starved cells led only to minimal changes in cell-cycle gene expression. Comparable effects were also observed on the protein level" This statement is only true at the mRNA level. Comparable effects were not observed at the protein level.
- Three biological replicates should be the minimum when performing qPCR analysis. The authors

should add more biological replicates to qPCR data with less than 3 biological replicates.

- "Idasanutlin-treatment also reduced cell-cycle gene mRNA (Fig. 4D) and protein (Fig.4E) expression similarly in wild-type and Sin3b^{-/-} cells and to a lesser extent in Lin37^{-/-}." This statement cannot be made for the protein levels as the western blot was not quantified.
- "Western Blot experiment were performed with at least two biological replicates with similar results." This statement in Fig. 4B legend is confusing. In (B), it is stated that only one clone of LIN37 KO was used.
- The authors should add an explanation of why only H3K27ac was measured and why H3K27me3 is measured.
- The authors should show statistical significance of the ChIP-qPCR data. The authors should also graph the average of all replicates done for the ChIP-qPCR experiment, instead of showing only one replicate. An additional biological replicate should be done as three biological replicates is conventional for this type of analysis.
- Fig 6A. legend is confusing. The authors are representing ChIP-qPCR data, yet they talk about mRNA expression. "mRNA expression was evaluated by RT-qPCR and compared to DMSO-treated cells for each respective line. One representative experiment with two technical replicates (mean values \pm SD) is shown. A biological replicate produced similar results. Data are presented as gene-set clusters (left) and individual genes (right)"
- Western blots in Figure 6. One biological replicate for each western blot is not enough for the conclusion drawn. The statement "A biological replicate for each Western blot experiment produced similar results" in the figure legend indicates that this experiment was done only once. All other figure legends are clear that at least two biological replicates were done
- "Upon doxorubicin treatment, repression of all analyzed cell-cycle genes did not change significantly or was even slightly, but significantly stronger in SIN3B^{-/-}". This sentence is confusing.
- "R_Core_Team" should be "R Core Team" in the methods and reference page.

o More information should be added regarding qPCR analysis (how mRNA % was calculated) in the methods section.

Reviewer #3:

Remarks to the Author:

In this manuscript, Barrett et al investigate the role of HDAC and Sin3 in repression of cell cycle genes by the DREAM complex. Earlier studies from several laboratories (for example, PMC152357) led to the model where E2F/RB recruits HDAC complexes to target promoters to actively repress their expression. The authors present a set of experiments in arrested HCT116 and C2C12 cells suggesting that the DREAM repression is largely independent of Sin3 and HDAC. The manuscript is well written, and the data are of excellent quality and well controlled. However, I feel that the manuscript falls a little bit short in advancing our understanding of how DREAM represses as the main conclusion is that it is not through Sin3 nor HDAC. Perhaps, delving deeper into what other chromatin modifying protein(s) are involved or exploring further the role of HDAC and Sin3 in repressing cell cycle genes independently of DREAM could improve the novelty and impact.

1) The authors show the SIN3B, SIN3A, and HDAC1 binding to cell-cycle genes in arrested HCT116 cells by ChIP. I am wondering whether DREAM plays a role in their recruitment. Can the authors perform these ChIP assays upon the loss of DREAM function?

2) The authors compare the cell cycle gene expression between Sin3 and DREAM knockouts in different settings. Could the differences be due to the recruitment of other E2Fs to these targets when DREAM is inactivated, which would not happen in Sin3 knockouts? Can the authors perform E2F ChIP experiments to address this point? Perhaps, the inclusion of a dominant negative E2F would help to confirm the results of DREAM inactivation.

We thank the reviewers and editors for their appreciation of our study and for their constructive criticism. We believe that addressing their comments substantially improved the quality of our manuscript.

Reviewer 1

Barrett et al describe a very thorough investigation of the roles played by Sin3B and Sin3A proteins, and by HDAC activity, in the regulation of cell cycle gene expression in human cell lines.

This is an important study. As the authors describe in their introduction, and in the discussion, many published studies have proposed that HDAC activities are recruited by RB and/or DREAM complexes to the promoters of cell-cycle regulated promoters, and that these proteins are important for the cyclic patterns of gene expression. But different studies have reported different results, and from reading the literature it is impossible to say with confidence exactly what the true roles of these proteins are.

Barrett et al have revisited this subject using gene targeting technology, and they are able to provide some clear answers. To some readers the results may seem disappointing because many of the results are negative. Barrett et al show that the genetic knockout of SIN3B did not cause the derepression of cell-cycle genes in non-proliferating HCT116 and C2C12 cells. The combined loss of SIN3A and SIN3B resulted in only a moderate upregulation of expression from several cell-cycle genes in arrested HCT116 cells. Supporting this Barrett et al show that HDAC inhibition does not induce a general upregulation of RB and DREAM target gene expression in arrested transformed or non-transformed cells. The results are especially convincing because Barrett et al are able to contrast these findings with the results that they detect in Lin37 deficient cells. Using compound mutants, they are able to make a good case that the lack of effects that they see with SIN3A/SIN3B/HDAC cannot be easily explained by redundancy. There are many interesting details but the main conclusion that I took from this work is that the claims that many studies have made about the roles of SIN3A, SIN3B and HDAC proteins at cell cycle genes have been overstated. It is very important that this is clarified in the literature. In contrast, the model that the authors recently proposed, that DREAM complexes position or stabilize nucleosomes over the transcription start site, seems more likely to be true. This model may apply to repression by both DREAM- and RB-complexes, but presently it is not clear (at least, not to me) how the authors will prove that this is the true mechanism of action.

I had just a couple of questions and a comment:

1. For the in silico analysis shown in Figure 1, what is the size of the promoter fragments that have been examined? This is an important detail and should be mentioned in the Figure legend. It would be interesting to know how the enrichment scores change if the analysis centers only on the DREAM or RB binding sites.

We are grateful for the reviewer's careful reading of the manuscript and grateful that they find our study important. We agree that it would be interesting to analyze binding of transcription factors and co-factors to cell-cycle genes depending on the distance to the MuvB and E2F/RB binding sites. Unfortunately, this is not possible with the analysis tools we found available. We contacted Luis del Peso Ovalle, the senior author of the paper that described the TFEA.ChIP tool and asked if there is a possibility to change the size of the analyzed promoter fragments relative to the TSS. His response was as follows:

"In response to your inquiry, TFEA.ChIP does not impose any specific distance cutoff for associating ChIP-seq peaks with genes. Instead, ChIP-seq peaks are linked to specific genes based on the evidence provided by the GeneHancer database. GeneHancer is a comprehensive database of human regulatory elements, including enhancers and promoters, along with their inferred target genes. The process involves mapping ChIP-seq peaks to the regulatory regions documented in the GeneHancer

database, and subsequently associating these peaks with the genes regulated by those specific regions. As a result, the distance between ChIP-seq peaks and their respective genes can vary significantly, and TFEA.ChIP does not apply any minimum or maximum distance threshold. This flexibility allows for more accurate gene associations, as it avoids the constraints of arbitrary distance thresholds. I trust that this explanation clarifies the matter and effectively addresses the reviewer's question. "

We abbreviated the explanation provided by Luis del Peso Ovalleto and added it to the Fig. 1 caption:

TFEA.ChIP maps ChIP-Seq peaks onto regulatory regions defined by the GeneHancer database (Fishilevich et al., 2017) and associates these peaks with the genes regulated by those regions.

Having seen the functional data, one wonders if the binding profile for the components of HDAC complexes can be separated in silico from the DREAM and E2F/RB proteins.

We agree that this analysis would be interesting in light of our functional data. However, it would require extensive ChIP-seq data sets for DREAM/E2F/RB components and HDAC complex proteins generated in the same cell line under comparable conditions, which are not available.

2. Cell cycle data is really important for these experiments. Because the authors are looking at cell cycle dependent transcription any changes in cell cycle position, even small changes, could alter patterns of expression. Perhaps the controls are in here and I just haven't searched carefully enough, but I couldn't find the data showing the knockouts had no effect on cell cycle distribution, and that the drug treatments had exactly the same effects (on DNA damage and cell cycle progression) that they have in the control cells. I assume that there are no major changes (because the main observation is the lack of changes in gene expression) but these controls need to be included.

We agree that these controls are important, and such data also add information, as we found that knockout/knockdown cells showed no or minimal changes in their ability to arrest. We added flow cytometry data to Fig. 2 (e, f), Fig. 4 (h), Fig. 5 (c, e), and Fig. 6 (d, e).

3. In the discussion please can the authors comment on whether they think that the conclusions that they draw extend to all cell types, and all physiological contexts, and whether they think that DREAM, MMB, and RB/E2F complexes have a single mechanism of action. There are hints in the text, but it might be helpful if the authors were to address these points directly.

We have attempted to address these points in the final paragraph of our discussion. Hopefully, it is clear that we still maintain the possibility of several mechanisms of gene repression, including indirect recruitment of HDACs, and that the significance of these mechanisms likely varies with physiological context.

“...our data suggest that HDAC activity is not generally required for cell-cycle gene repression during reversible cell-cycle arrest and that alternative mechanisms like RB-dependent inhibition of activator E2Fs (Helin et al., 1993; Hiebert et al., 1992), stabilizing nucleosomes by DREAM (Asthana et al., 2022), and potentially recruitment of other chromatin-modifying or nucleosome-remodeling proteins are sufficient to induce robust repression. However, HDAC activity may be involved in fine-tuning the activity of subsets of cell-cycle genes under specific physiological conditions (Sanidas et al., 2019), and by influencing chromatin structure, DREAM may play an indirect role in HDAC recruitment. Furthermore, in contrast to reversible cell-cycle arrest, HDAC activity may be required for the permanent silencing of cell-cycle genes during terminal cell-cycle exit. This topic has yet to be well-studied, but several transcriptome data sets derived from

terminally differentiated cells negative for HDAC1/2, or treated with HDAC inhibitors, are available and suggest that HDAC activity is also not generally required in this context (Montgomery et al., 2007; Rai et al., 2008). Taken together, it remains to be elucidated in which biological contexts SIN3 proteins and HDAC activity substantially contribute to cell-cycle gene regulation.”

Reviewer 2

Noteworthy results

o Barrett et al. show that SIN3B is not required for the inhibition of some cell cycle genes induced by DNA damage or P53 activation.

o Barrett et al. show that SIN3B is required for some cell cycle gene repression when grown in serum starved conditions. Inducing DNA damage in LIN37 KO also leads to derepression of cell cycle genes. They also show that LIN37 does CoIP with a complex containing HDAC activity in serum starved conditions.

o Barrett et al. show that LIN37 KO in mouse cell line C2C12 does not phenocopy SIN3B KO at the mRNA level but not the protein level, in response to serum starvation.

o Barrett et al. show that SIN3A function is partially redundant with SIN3B function independent of RB or LIN37.

o Barrett et al. show that Class I HDACs are not involved in the repressive mechanism of LIN37 on cell cycle genes.

The work is important and will advance the scientific knowledge of the mechanism in which the DREAM complex regulates cell cycle genes.

Overall, the work somewhat supports the conclusions drawn. I have noted below where the data did not support the conclusion drawn and where additional experiments should be done:

Overall note: Three biological replicates should be done when appropriate. Average of all biological replicates should be presented when representing data as a graph.

We thank the reviewer for their critical reading of the manuscript, and in particular, for indicating which conclusions need to be better supported by additional data and analysis.

We agree that including more replicates improves the quality of the data sets. As suggested, we added additional replicates to the experiments described in Fig. 2 (b, c), Fig. 3 (c), Fig. 4 (b, d, f), Fig. 5 (b, d), and Fig. 7 (a).

• *“Gene repression was impaired in LIN37^{-/-} and RB^{-/-} cells, and we detected a stronger loss of repression for G2/M genes in LIN37^{-/-} cells and for G1/S genes in RB^{-/-} cells” Cannot make this statement as statistical significance between LIN37 KO / RB KO cells and WT cells was not done.*

We removed the clause “and we detected a stronger loss of repression for G2/M genes in LIN37^{-/-} cells and for G1/S genes in RB^{-/-} cells.” This conclusion is not particularly relevant for the comparison of cells with or without SIN3B.

• *Importantly, combined loss of SIN3B and RB did not reflect the almost complete loss of repression observed in LIN37^{-/-};RB^{-/-} cells” Cannot make this statement as statistical significance was not done. In any case, gene expression of MCM5 and ORC1 in the cells with dual SIN3B and RB KO is not very different from LIN37 RB dual KO cells.*

We removed the sentence.

- *Since the authors are making conclusion by western blot analysis, quantification of bands should be done for Fig. 2D.*

Considering that our conclusions are qualitative, we are hesitant to add quantification of Western blot band intensities. To the contrary, whereas we are careful to make rather conservative conclusions, we have concerns that inclusion of a quantification may entice readers to overinterpret subtle differences in the bands.

- *Fig. 2E. The X and Y axis are relative to what? This information should be added to the legend*

We changed the title of the axis to “HDAC I/II activity” and “IP eluant in reaction”.

- *“mRNA levels of all analyzed genes were strongly reduced in starved wild-type cells” The authors need to be clear that the statistical significance was only measured at 120 hrs, thus this statement is misleading. Alternatively, you can measure the statistical significance of all timepoints. This is important because later experiments use different starvation times. Additionally, the *** indicating significance is very small and cannot be read.*

We now show a serum-starvation experiment with just two time points (48h, 96h) but three biological replicates in Fig. 4b, and we moved the time-course experiment to Suppl. Fig. 1. We clarify in the caption of Suppl. Fig. 1 that the significance was only calculated for the 120 hr time point, and we increased the size of the graphs in the figure to be more legible.

- *Fig 3B. The authors should add a sentence on what mRNA % means, as fold change is the conventional way to represent qPCR data.*

We now only show log2 fold changes in mRNA expression experiments in the main figures. We clarified what mRNA [%] means in the caption of the new Suppl. Fig 1a.

- *Fig. 3C. The WB is confusing and not consistent with the rest of the paper. The loading control should be at the bottom.*

For this experiment, we were required to run the samples on two different gels and to probe two different blots (i.e. WT and SIN3B knockouts on left gel in panel and LIN37 and double knockouts on the right gel). We found it important to show that there is a clear difference in LIN37 abundance between the wild-type and LIN37 knockout cells by probing the blots with both the LIN37 and control actin antibodies. Otherwise, we would have to show just an empty blot for the LIN37 antibody with knockout cells. Thus, we would like to keep the figure as presented, in which the long and short LIN37 exposures are kept together, even if this means the loading control position seems out of order.

- *Fig. 3D panel 1. Y axis is different from the rest of the panel.*

The figure has been changed and is now consistent with the other similar panels.

- *“Additionally, Palbociclib treatment led to comparable repression of DREAM targets in wild-type and SIN3B^{-/-} cells on the protein level, while increased expression was detected in LIN37^{-/-} and SIN3B^{-/-};LIN37^{-/-} cells” This statement is not true as the amount of Survivin and B-MYB at 0 hr is different from WT and SIN3B KO, therefore the change is due to the KO and not treatment. Additionally, the authors needs to quantify western blots when using western blots to quantify expression levels.*

- *“In contrast, addition of Palbociclib to LIN37-negative serum-starved cells led only to minimal changes in cell-cycle gene expression. Comparable effects were also observed on the protein level” This statement is only true at the mRNA level. Comparable effects were not observed at the protein level.*

In order to keep the conclusion focused on the most important and conservative interpretation of the Western blot, we changed the statement to read as follows:

Palbociclib treatment for 48h led to a reduction of B-MYB and survivin proteins in SIN3B^{-/-} cells comparable to the wild-type cells, while protein levels remained elevated in LIN37^{-/-} or SIN3B^{-/-};LIN37^{-/-} clones (Fig. 4f, g).

- *Three biological replicates should be the minimum when performing qPCR analysis. The authors should add more biological replicates to qPCR data with less than 3 biological replicates.*

As suggested and indicated above, we added additional replicates to these experiments. The qPCR experiments now contain at least three biological replicates. We note that in some experiments, we use two independent clones and perform two independent experiments with each; we consider these replicates as four biological replicates. The additional data did not change our conclusion.

- *“Idasanutlin-treatment also reduced cell-cycle gene mRNA (Fig. 4D) and protein (Fig.4E) expression similarly in wild-type and Sin3b^{-/-} cells and to a lesser extent in Lin37^{-/-}” This statement cannot be made for the protein levels as the western blot was not quantified.*

As described above, we are not inclined to include quantification of Western blot band intensities out of concern that such analysis will lead to overinterpretation of our data. Alternatively, we have restricted our conclusions to conservative qualitative descriptions (e.g. “reduced similarly” and “to a lesser extent”) that we believe are obvious in the blots and sufficiently support our hypotheses.

- *“Western Blot experiment were performed with at least two biological replicates with similar results.” This statement in Fig. 4B legend is confusing. In (B), it is stated that only one clone of LIN37 KO was used.*

We added more replicates and an additional LIN37 clone to the data set.

- *The authors should add an explanation of why only H3K27ac was measured and why H3K27me3 is measured.*

For simplicity, we removed the H3K27me3 data and added the following explanation for why H3K27ac was measured:

As an example of a canonically activating histone mark, we analyzed whether H3K27 acetylation on cell-cycle gene promoters changes when arrested with Idasanutlin and treated with Romidepsin.

- *The authors should show statistical significance of the ChIP-qPCR data. The authors should also graph the average of all replicates done for the ChIP-qPCR experiment, instead of showing only one replicate. An additional biological replicate should be done as three biological replicates is conventional for this type of analysis.*

We have added data such that all ChIP-qPCR experiments now contain averages and significances of three biological replicates (Fig. 3c, 3e, 3g, and Fig. 7a).

• Fig 6A. legend is confusing. The authors are representing ChIP-qPCR data, yet they talk about mRNA expression. “mRNA expression was evaluated by RT-qPCR and compared to DMSO-treated cells for each respective line. One representative experiment with two technical replicates (mean values \pm SD) is shown. A biological replicate produced similar results. Data are presented as gene-set clusters (left) and individual genes (right)”

The figure caption (now Fig. 7) has been updated to read as follows:

(A) H3K27 acetylation at cell-cycle gene promoters in HCT116 cells was analyzed by ChIP-qPCR. Three biological replicates with two technical replicates (mean values \pm SEM) are shown. For data shown in (b, d, f, h), data are presented as gene-set clusters (left) and individual genes (right) in: (B) HCT116, (D) A549, (F) C2C12, and (H) BJ-hTERT cells.

• Western blots in Figure 6. One biological replicate for each western blot is not enough for the conclusion drawn. The statement “A biological replicate for each Western blot experiment produced similar results” in the figure legend indicates that this experiment was done only once. All other figure legends are clear that at least two biological replicates were done.

We agree that this statement was confusing and changed the figure caption to read as follows:

An additional biological replicate for each Western blot experiment produced similar results.

• “Upon doxorubicin treatment, repression of all analyzed cell-cycle genes did not change significantly or was even slightly, but significantly stronger in SIN3B^{-/-}”. This sentence is confusing.

We simplified the sentence to read as follows:

Upon doxorubicin treatment, repression of all analyzed cell-cycle genes did not change significantly or was slightly stronger in SIN3B^{-/-} cells.

• “R_Core_Team” should be “R Core Team” in the methods and reference page.

We fixed this typo.

More information should be added regarding qPCR analysis (how mRNA % was calculated) in the methods section.

As described above, we added a description to how mRNA % was calculated to the appropriate figure caption.

Reviewer 3

In this manuscript, Barrett et al investigate the role of HDAC and Sin3 in repression of cell cycle genes by the DREAM complex. Earlier studies from several laboratories (for example, PMC152357) led to the model where E2F/RB recruits HDAC complexes to target promoters to actively repress their expression. The authors present a set of experiments in arrested HCT116 and C2C12 cells suggesting that the DREAM repression is largely independent of Sin3 and HDAC. The manuscript is well written, and the data are of excellent quality and well controlled. However, I feel that the manuscript falls a little bit short in advancing our understanding of how DREAM represses as the main conclusion is that it is not through Sin3 nor HDAC.

Perhaps, delving deeper into what other chromatin modifying protein(s) are involved or exploring further the role of HDAC and Sin3 in repressing cell cycle genes independently of DREAM could improve the novelty and impact.

We agree with the first two reviewers, who argue the importance of this work and that in fact, by concluding that SIN3 and HDAC proteins are not essential for the mechanism of DREAM repression, we have significantly advanced our understanding of DREAM. We believe that particularly because of conflicting literature, it is an important advance to demonstrate that SIN3 proteins and HDAC activity are not essential for DREAM-dependent gene repression.

1) The authors show the SIN3B, SIN3A, and HDAC1 binding to cell-cycle genes in arrested HCT116 cells by ChIP. I am wondering whether DREAM plays a role in their recruitment. Can the authors perform these ChIP assays upon the loss of DREAM function?

We agree that this is an important question to address. However, with existing cell lines, it was not feasible for several reasons to run ChIP assays that specifically probe loss of DREAM function. First, while the LIN37-negative cells used here lose DREAM function in gene repression, the remaining DREAM complex is still recruited to promoters and could recruit corepressors. Studying loss of DREAM through knockout of other proteins in the complex would either similarly leave the complex on promoters (e.g. p130) or disrupt the activator MuvB complexes, which would lead to defects in proliferation (e.g. LIN9, LIN54).

To circumvent these challenges, we used the CRISPR approach and created two cell lines with mutations in the CHR elements of the *BUB1* or *CCNB2* promoters. These cells proliferate normally but cannot recruit MuvB to only these particular promoters. This experimental design allowed us to analyze binding of SIN3 proteins and HDAC1 in the absence of DREAM at these promoters as suggested. As shown in Fig. 3e, binding of SIN3 and HDAC to the *BUB1* promoter in the absence of DREAM is reduced, but still clearly above background. In contrast, binding of SIN3/HDAC to the *CCNB2* promoter is reduced to almost background levels when DREAM cannot bind anymore. Thus, we conclude that DREAM supports binding of SIN3/HDAC in a promoter-specific manner, but that these proteins can still bind to at least a subset of cell-cycle genes when DREAM binding is abrogated.

2) The authors compare the cell cycle gene expression between Sin3 and DREAM knockouts in different settings. Could the differences be due to the recruitment of other E2Fs to these targets when DREAM is inactivated, which would not happen in Sin3 knockouts? Can the authors perform E2F ChIP experiments to address this point? Perhaps, the inclusion of a dominant negative E2F would help to confirm the results of DREAM inactivation.

Our studies are using LIN37 knockout to study loss of DREAM function, and in LIN37^{-/-} cells, the DREAM complex, including E2F4, is still recruited to cell-cycle gene promoters (see Mages et al., eLife, 2017; Uxa et al., NAR, 2019). Moreover, upon induction of cell-cycle arrest, activator E2F protein levels are downregulated. For these reasons, we have no expectation that other E2Fs are being recruited when cell-cycle arrest is induced in our experiments.

Reviewers' Comments:

Reviewer #1:

Remarks to the Author:

The authors have addressed all of the points that I raised in my review of this manuscript and I recommend that this work should be published. As described in the initial review, several different models have been proposed for the mechanism of action of DREAM and this manuscript has done a very impressive job of testing several of the claims. This is the first study to do this in any meaningful way. The experiments have been done to a high standard, the results are clear, and the data has been interpreted carefully. In my opinion, this study represents an important contribution to the field.

Reviewer #2:

Remarks to the Author:

Noteworthy results

- o Barrett et al. show that SIN3B is not required for the inhibition of some cell cycle genes induced by DNA

damage or P53 activation.

- o Barrett et al. show that SIN3B is required for some cell cycle gene repression when grown in serum starved conditions. Inducing DNA damage in LIN37 KO also leads to derepression of cell cycle genes. They also show that LIN37 does CoIP with a complex containing HDAC activity in serum starved conditions.

- o Barrett et al. show that LIN37 KO in mouse cell line C2C12 does not phenocopy SIN3B KO at the mRNA

level but not the protein level, in response to serum starvation.

- o Barrett et al. show that SIN3A function is partially redundant with SIN3B function independent of RB or LIN37.

- o Barrett et al. show that Class I HDACs are not involved in the repressive mechanism of LIN37 on cell cycle genes.

The work is important and will advance the scientific knowledge of the mechanism in which the DREAM complex regulates cell cycle genes.

Overall, they have responded to all my comments and the revised manuscript strongly supports the conclusions drawn.

Reviewer #3:

Remarks to the Author:

The authors adequately addressed my concerns in this revision and I am supportive of publication.